**Development and testing of scenarios for implementing land use and land cover changes during the Holocene in Earth System Model Experiments**

Sandy P. Harrison[1], Marie-José Gaillard[2], Benjamin D. Stocker[3,4], Marc Vander Linden[5], Kees Klein Goldewijk[6,7], Oliver Boles[8], Pascale Braconnot[9], Andria Dawson[10], Etienne Fluet-Chouinard[11], Jed O. Kaplan[12,13], Thomas Kastner[14], Francesco S.R. Pausata[15], Erick Robinson[16], Nicki J. Whitehouse[17], Marco Madella[18,19, 20], Kathleen D. Morrison[8]

Ms for Geoscientific Model Development (PMIP special issue)

1: Department of Geography and Environmental Science, University of Reading, Reading, UK
2: Department of Biology and Environmental Science, Linnaeus University, Kalmar, Sweden
3: Ecological and Forestry Applications Research Centre, Cerdanyola del Vallès, Spain
4: Department of Earth System Science, Stanford University, Stanford, CA 94305, USA
5: Department of Archaeology, University of Cambridge, UK
6: PBL Netherlands Environmental Assessment Agency, The Hague, The Netherlands
7: Copernicus Institute of Sustainable Development, Utrecht University, The Netherlands
8: University Museum of Archaeology & Anthropology, University of Pennsylvania, Philadelphia, USA
9: Laboratoire des Sciences du Climat et de l'Environnement, Gif-sur-Yvette, France
10: Department of General Education, Mount Royal University, Calgary, Canada
11: Department of Earth System Science, Stanford University, California, USA
12: Department of Earth Sciences, University of Hong Kong, Hong Kong
13: Institute of Geography, University of Augsburg, Augsburg, Germany
14: Senckenberg Biodiversity and Climate Research Centre, Frankfurt am Main, Germany
15: Centre ESCER, Department of Earth and Atmospheric Sciences, University of Quebec in Montreal, Montreal, Canada
16: Department of Anthropology, University of Wyoming, Laramie, Wyoming, USA
17: School of Geography, Earth and Environmental Science, University of Plymouth, Plymouth, UK
18: Department of Humanities (CaSEs), University Pompeu Fabra, Barcelona, Spain
19: ICREA Passeig Lluís Companys 23 08010 Barcelona, Spain
20: School of Geography, Archaeology and Environmental Studies, University of Witwatersrand, Johannesburg, South Africa

**Abstract**: Anthropogenic changes in land use and land cover (LULC) during the pre-industrial Holocene could have affected regional and global climate. Existing scenarios of LULC changes during the Holocene are based on relatively simple assumptions and highly uncertain estimates of population changes through time. Archaeological and palaeoenvironmental reconstructions have the potential to refine these assumptions and estimates. The Past Global Changes (PAGES) LandCover6k initiative is working towards improved reconstructions of LULC globally. In this paper, we document the types of archaeological data that are being collated and how they will be used to improve LULC reconstructions. Given the large methodological uncertainties involved, both in reconstructing LULC from the archaeological data and in implementing these reconstructions into global scenarios of LULC, we propose a protocol to evaluate the revised scenarios using independent pollen-based reconstructions of land cover and climate. Further evaluation of the revised scenarios involves carbon-cycle model simulations to determine whether the LULC reconstructions are consistent with constraints provided by ice-core records of $CO_2$ evolution and modern-day LULC. Finally, the protocol outlines how the improved LULC reconstructions will be used in palaeoclimate simulations in the Palaeoclimate Modelling Intercomparison Project to quantify the magnitude of anthropogenic impacts on climate through time and ultimately to improve the realism of Holocene climate simulations.

## 1 Introduction and Motivation

Today, ca 10% the ice-free land surface is estimated to be intensively managed and much of the reminder is under less intense anthropogenic use or influenced by human activities (Arneth et al., 2019). Substantial transformations of natural ecosystems by humans began with the geographically diachronous shift from hunting and gathering characteristic of the Mesolithic to cultivation and more permanent settlement during the Neolithic period (Mazoyer and Roudart, 2006; Zohary et al., 2012; Tauger, 2013; Maezumi et al. 2018), although there is controversy about the relative importance of climate changes and human impact on landscape development both during and since that time. Resolving the uncertainty about the extent and timing of land use is important because changes in land cover as a result of land use (land use land cover: LULC) have the potential to impact climate and the carbon cycle. Direct climate impacts occur through changes in the surface-energy budget resulting from modifications of surface albedo, evapotranspiration, and canopy structure (biophysical impacts, e.g. Pongratz et al., 2010; Myhre et al., 2013; Perugini et al., 2017). LULC affects the carbon cycle through modifications in vegetation and soil carbon storage (biogeochemical impacts, e.g. Pongratz et al., 2010; Mahowald et al., 2017) and turnover times, which changes the C sink/source capacity of the terrestrial biosphere. LULC changes have contributed substantially to the increase in atmospheric greenhouse gases during the industrial period (Le Quéré et al., 2018). It has been suggested that greenhouse gas emissions associated with Neolithic LULC changes were sufficiently large to offset climate cooling after the Mid-Holocene (the overdue-glaciation hypothesis: Ruddiman 2003). Although this has been challenged for several reasons, including inconsistency with the land carbon balance derived from ice-core and peat records (e.g. Joos et al., 2004; Kaplan et al., 2011; Singarayer et al., 2011; Mitchell et al., 2013; Stocker et al. 2017), a LULC impact on climate in more recent millennia appears more plausible.

Climate model simulations have shown that LULC changes have discernible impacts on climate, both in regions with large prescribed changes in LULC and in teleconnected regions with no major local human activity (Vavrus et al., 2008; Pongratz et al., 2010; He et al., 2014; Smith et al., 2016). At the global scale, the biogeophysical effects of the accumulated LULC change during the Holocene which resulted in reconstructed land cover patterns in 1850CE have been estimated to cause a slight cooling (0.17 °C) that is offset by the biogeochemical warming (0.9 °C), giving a net global warming (0.73 °C) (He et al., 2014). However, in these simulations, biophysical and biogeochemical effects were of comparable magnitude in the most intensively altered landscapes of Europe, Asia, and North America (He et al., 2014). Using parallel simulations, with and without LULC changes, Smith et al. (2016) showed that detectable temperature changes due to LULC could have occurred as early as 7000 years ago (7ka BP) in summer and throughout the year by 3ka BP. All of these conclusions, however, are obviously contingent on the imposed LULC forcing, which is highly uncertain.

There have been several attempts to map LULC changes through time (e.g. Ramankutty and Foley, 1999; Pongratz et al., 2008; Kaplan et al., 2011; Klein Goldewijk et al. 2011; Klein Goldewijk et al. 2017a, b). All of these reconstructions assume that anthropogenic land use is a function of population density and the suitability of land for crops and/or pasture. They then use estimates of regional population trends through time in combination with assumptions about per-capita land use and spatial land use allocation schemes to estimate anthropogenic changes in LULC across time and space. However, differences in the underlying assumptions about land-use per capita, which are generalized from limited and often site-specific data, have resulted in large differences in the final reconstructions (Gaillard et al., 2010; Kaplan et al., 2017). Hence, there are still very large uncertainties about the timing and magnitude of LULC changes, both at a global and at a regional scale (Figure 1).

There is a wealth of archaeological, historical and palaeo-vegetation data that could be used to

improve the relatively simple rules used to generate global LULC reconstructions. For example, settlement density and numbers of radiocarbon-dated artifacts can be used to infer population sizes and their temporal dynamics (Rick, 1987; Williams, 2012; Silva and Vander Linden, 2017). Carbonised and waterlogged plant remains and animal bones can be used to infer the nature of agriculture at a site, although their presence provides no quantitative information about the area under cultivation (Wright, 2003; Lyman 2008; Orton et al., 2016). Although the record of LULC is likely to be patchy and incomplete, because of preservation and sampling issues, systematic use of archaeological data is one important way to improve current LULC scenarios.

The Past Global Change (PAGES, http://www.pastglobalchanges.org/) LandCover6k Working Group (http://pastglobalchanges.org/ini/wg/landcover6k) is currently working to develop a rigorous and robust approach to provide data and data products that can be used to inform reconstructions of LULC (Gaillard et al., 2018). LULC changes are taken into account in simulations currently being made in the current phase of the Coupled Model Intercomparison Project (CMIP6) for the historic period and the future scenario runs (Eyring et al., 2016). They are also included in simulations of the past millennium (Jungclaus et al., 2017), in order to ensure that these runs mesh seamlessly with the historic simulations. However, the Land Use Harmonisation data set (LUH2: Hurtt et al., 2017) only extend back to 850 CE and thus LULC changes are currently not included in the CMIP6 palaeoclimate simulations, including mid-Holocene simulations, that are used as a test of how well state-of-the-art climate models reproduce large climate changes. In this paper, we discuss how archaeological data will be used to improve global LULC reconstructions for the Holocene. Given that there are large uncertainties associated with the primary data and further uncertainties may be introduced when this information is used to modify existing LULC scenarios, we outline a series of tests that will be used to evaluate whether the revised scenarios are consistent with the changes implied by independent pollen-based reconstructions of land cover and whether they produce more realistic estimates of both carbon cycle and climate change. Finally, we present a protocol for implementing LULC in Earth System Model simulations to be carried out in the current phase of the Palaeoclimate Modelling Intercomparison Project (PMIP: Otto-Bleisner et al., 2017; Kageyama et al., 2018). However, the data sets and protocol will also be useful in later phases of other CMIP projects, including the Land Use Model Intercomparison Project (LUMIP) and the Land Surface, Snow and Soil Moisture Model Intercomparison Project (LS3MIP) (Lawrence et al., 2016; van den Hurk et al., 2016).

**2 LandCover6k Methodology**

The primary source of information about human exploitation of the landscape comes from archaeological data. In general, these data are site specific and spatiotemporal coverage is often patchy, and the types and quality of evidence available vary between sites and regions. Generalising from site-specific data to landscape or regional scales involves making assumptions about human behaviour and cultural practices. Because of the inherent uncertainties, we advocate an iterative approach to incorporate archaeological data into LULC scenarios in LandCover6k (Fig. 2). We propose to revise the LULC scenario by incorporation of diverse archaeological inputs (Fig. 2, phase 1; see Sections 3 and 4) and to test the revised LULC scenarios for their plausibility and consistency with other lines of evidence (Fig. 2, phase 2 with iterative testing; see Sections 5-7). As a first test, the revised LULC scenarios of the extent of cropland and grazing land through time will be compared with independent data on land-cover changes, specifically pollen-based reconstructions of the extent of open land (see e.g. Trondman et al., 2015; Kaplan et al., 2017) (Section 5). Further testing the LULC scenarios involve sensitivity tests using global climate models (Section 6) and global vegetation-carbon cycle models (Section 7). While the computational cost of the climate simulations can be minimized using equilibrium time-slice simulations, the carbon cycle constraint relies on

transient simulations, but may be derived from uncoupled, land-only simulations. Simulated climates at key times can be evaluated against reconstructions of climate variables (e.g. Bartlein et al., 2011) (Section 6). The parallel evolution of $CO_2$ and its isotopic composition ($\delta^{13}C$) can be used to derive the carbon balance of the terrestrial biosphere and the ocean separately (Elsig et al., 2009) and, in combination with estimates for other contributors to land carbon changes such as C sequestration by peat buildup, provides a strong constraint on the evolution of LULC through time. An under- or over-prediction of anthropogenic LULC-related $CO_2$ emissions during a specific interval results in consequences for the dynamics of the atmospheric greenhouse gas burden in subsequent times (Stocker et al., 2017) (Section 7). Thus, these tests can be used to identify issues in the original archaeological datasets and/or the way these data were incorporated into the LULC scenarios that require further refinement. Phase 3 of the protocol (Fig. 2) proposes specific implementation of the revised LULC in Earth System Model simulations (Section 8).

## 3 Archaeological data inputs

LandCover6k is creating a number of products that will be used to improve the LULC scenarios (Figure 2). Here, we summarise the important features of these data products before showing how they will be incorporated within a scenario-development framework.

### 3.1 Population dynamics from $^{14}C$ data

Radiocarbon is the most routinely used absolute dating technique in archaeology, especially for the Holocene. Many thousands of radiocarbon dates are available from the archaeological literature. A number of regional and pan-regional initiatives are compiling these records through exhaustive survey of the archaeological literature (e.g. the Canadian Archaeological Radiocarbon Database: https://www.canadianarchaeology.ca/). Statistical approaches, such as summed probability distributions (SPDs), can then be used to infer past demographic fluctuations from these compilations (Figure 3). This method assumes that the more people there were, the more remains of their various activities they left behind, and that this is directly reflected in the number of samples excavated and dated (Rick, 1987: Robinson et al., 2019). There are biases that could affect the expected one-to-one relationship between number of people and number of radiocarbon dates on archaeological material, including lack of uniform sampling through time and space caused by different archaeological research interests and traditions in different regions and increased preservation issues with increasing age, but these can be minimised through auditing the datasets. Assessment of the robustness of population reconstructions through time can be made statistically, by comparing a null hypothesis of demographic growth constructed from an exponential fit to the data with the actual record of number of dates through time (Shennan et al., 2013; Timpson et al., 2014). Mathematical simulations show that the method is relatively robust for large sample sizes (Williams, 2012). Radiocarbon dates have been successfully used in several regions to identify population fluctuations associated with the introduction of farming and subsequent changes in farming regimes (western Europe: Shennan et al., 2013; Wyoming: Zahid et al., 2016; South Korea: Oh et al., 2017; see also Freeman et al., 2018) as well as climatic oscillations (Ireland: Whitehouse et al., 2014; Japan: Crema et al., 2016).

### 3.2 Date of first agriculture

Radiocarbon dates can also be used to track the timing and process of dispersal events, such as the diffusion of plant and animal domesticates from their initial centres of domestication. Since the distribution of samples is often patchy, geostatistical techniques such as kriging and splines are used to spatially interpolate the information in order to provide quantitative estimates of the timing of spread. Work carried out in Europe (Bocquet-Appel et al., 2009), Asia (Silva et al., 2015), and Africa (Russell et al., 2014) demonstrates that there are different rates of diffusion even within a region, reflecting the possible impact of natural features (e.g. waterways, elevation, ecology) on diffusion

rates (Davison et al., 2006; Silva and Steele, 2014). Numerous studies provide robust local estimates
for the earliest regional occurrence of agriculture and these are being synthesized to provide a global
product within LandCover6k (Figure 2).
**3.3 Global land-use and livestock maps**
Maps of the distribution of archaeological sites or of areas linked to a given food production system
have been produced for individual site catchments or small regions (e.g. Zimmermann et al., 2009;
Barton et al., 2010; Kay et al., in press). LandCover6k is developing global land-use maps for specific
time windows, using a global hierarchical classification of land-use categories (Morrison et al., 2018)
based on land-use types that are widely recognised from the archaeological record. At the highest
level, the maps distinguish between areas where there is no (or only limited) evidence of land use,
and areas characterized by hunting/foraging/fishing activities, pastoralism, agriculture, and
urban/extractive land use (Fig. 4). Except in the cases where land use is minimal (no human land use,
extensive/minimal land use), further distinctions are subsequently made to encompass the diversity
of land-use activities in each land-use type (Fig. 4). A third level of distinction is made in the case of
two categories (agroforestry, wet cultivation) where there are very different levels of intervention in
different regions. Explanations of this terminology are given in Morrison et al. (2018). The
LandCover6k land-use maps (see e.g. Fig. 5) will be based on different methods ranging from kernel-
density estimates to expert assessments depending on the quality and quantity of the archaeological
information available from different regions.
There is considerable variation in how intensely land is used both for crops and for grazing within
broad land-use categories both geographically and through time (Ford and Clarke, 2015; Styring et
al., 2017). Maps of land-use types do not provide direct information on the intensity of farming
practices or how they translate into per-capita land use. Archaeological data about agricultural yields,
combined with information from analogous contemporary cultures, historical information (e.g.
Pongratz et al., 2008) and theoretical estimates of land use required to meet dietary and energy
requirements (e.g. Hughes et al., 2018), can be used to provide regional estimates of per-capita land
use for specific land-use categories. LandCover6k will synthesise this information to allow regionally
specific estimates of per-capita land use to be derived from the global land-use maps.
Information about the extent of grazing land is an important input to LULC scenarios but, from a
carbon-cycle modelling perspective, the amount of biomass removed by grazing is also a key
parameter. Biomass loss varies not only with population size but also with the type of animal being
reared (Herrero et al., 2013; Phelps & Kaplan, 2017) and thus information about what animals were
present at a given location and estimates of population sizes are needed for LULC scenarios. Although
the conditions of bone preservation vary across the globe due to factors such as soil acidity, animal
bones are routinely excavated (Lyman, 2008; Reitz & Wing, 2008). Morphometric analysis of bones,
along with collateral information such as age-related culling patterns, make it possible to determine
whether these are the remains of domesticated species. We thus have a relatively precise idea of when
livestock were introduced into a region and what types of animal were being reared at a given time,
and can also make informed estimates of population size. Although the level of detail will vary
geographically, this information can be used to produce global livestock maps.
The harvesting of wood for domestic fires, building, and for industrial activities such as
transportation, pottery-making and metallurgy is an important aspect of human exploitation of the
landscape in the pre-industrial period (McGrath et al., 2015). It has been argued that even Mesolithic
hunter-gatherer communities shaped their environment through wood harvesting (Bishop et al.,
2015). Approaches have been developed to quantifying the wood harvest associated with
archaeological settlements at specific times based on the evidence of types of wood use, household
energy requirements, population size, and calorific value of the wood used (see e.g. Marston, 2009;
Janssen et al., 2017). However, quantitative information on ancient technology and lifestyle is sparse
and direct estimates of the amount of wood harvest through time are likely to remain highly uncertain
(Marston et al., 2017; Veal, 2017). Nevertheless, by combining modelling approaches with improved
estimates of population size should allow changes in wood harvesting to be taken into account in
LULC scenarios.
**4. Incorporation of archaeological data in LULC scenarios**
The existing LULC scenarios are substantially dependent on historical regional population estimates
at key times, which are then linearly interpolated to provide a year-by-year estimate of population.
Estimates of regional population growth based on suitably-screened $^{14}$C data can be used to modify
existing population growth curves (Figure 6), both in terms of establishing the initial date of human
presence and by modifying a linear growth curve to allow for intervals of population growth and
decline.
Information on the timing of the first appearance of agriculture at specific locations can be used to
constrain the temporal record of LULC changes in the scenarios. This information can also be used
to allocate LULC changes geographically across regions (Figure 6). Global land-use maps can be
used to identify areas where there was no permanent agricultural activity at a given time (e.g. either
unsettled areas or areas occupied by hunter-gatherer communities) and provide a further constraint
on the geographic extent of LULC changes (Figure 6). The type of agriculture, including whether the
region was predominantly used for tree or annual crops or for pasture, modifies the area of open land
specified in the scenarios. Information on the extent of rain-fed versus irrigated agriculture, as
indicated by the presence of irrigation structures associated with archaeological sites, can also be used
to refine the distribution of these classes in the LULC scenarios. Per-capita land-use estimates and
their changes through time (see e.g. Hughes et al., 2018; Weiberg et al., 2019) provide a further
refinement of the LULC scenarios, allowing a better characterization of the distinction between e.g.
areas given over to extensive versus intensive animal production (rangeland versus pasture in the
HYDE 3.2 terminology). There will remain areas of the world for which this kind of fine-grained
information is not available. Nevertheless, by incorporating information where this exists, the
LandCover6k products will contribute to a systematic refinement of LULC scenarios. Iterative testing
of the revised scenarios will ensure that they are robust.
**5. Using pollen-based reconstructions of land cover changes to evaluate LULC scenarios**
Pollen-based vegetation reconstructions can be used to corroborate archaeological information on the
date of first agriculture from the appearance of cereals and agricultural weeds. These reconstructions
can also be used to test the LULC reconstructions, either using relative changes in forest cover or
reconstructions of the area occupied by different land cover types. LandCover6k uses the REVEALS
model (Sugita, 2007) to estimate vegetation cover from fossil pollen assemblages. The REVEALS
model predicts the relationship between pollen deposition in large lakes and the abundance of
individual plant taxa in the surrounding vegetation at a large spatial scale (ca. 100 km x 100 km;
Hellman et al., 2008a, b) using models of pollen dispersal and deposition. REVEALS can also be
used with pollen records from multiple small lakes or peat bogs (Trondman et al., 2016) although this
results in larger uncertainties in the estimated area occupied by individual taxa. The estimates
obtained for individual taxa are summed to produce estimates of the area occupied by either plant

functional (e.g. summer-green trees, evergreen trees) or land cover (e.g. open land, grazing land, cropland) types.

The geographic distribution of pollen records is uneven. There are also many areas of the world where environments that preserve pollen (i.e. lakes, bogs, forest hollows) are sparse. Site-based reconstructions of land cover are therefore interpolated statistically to produce spatially continuous reconstructions (Nielsen et al., 2012; Pirzamanbein et al., 2014; Pirzamanbein et al., 2018). LandCover6k uses a 1˚ resolution grid and all available pollen records in each grid cell to produce an estimate of land cover per grid cell through time. The more pollen records per grid cell and pollen counts per time window, the smaller the estimated error on the land-cover reconstruction. The uncertainties on the pollen-based REVEALS estimates are partly expressed by their standard errors (SEs). These SEs take into account the SE on the relative pollen productivity (RPP) of each plant taxon included in the REVEALS reconstruction and the variability between the site-specific REVEALS estimates (e.g. Trondman et al., 2015). These uncertainties on the pollen-based land cover are considered when these reconstructions are compared with LULC scenarios (Kaplan et al., 2017).

The REVEALS approach has already been used to produce gridded reconstructions of changes in the amount of open land through time across the northern extratropics (Figure 7; Dawson et al., 2018) These reconstructions provide mean plant cover for time slices of 500 years through the Holocene until 0.7ka BP, and three historical time windows (modern–0.1ka BP, 0.1–0.35ka BP, and 0.35–0.7ka BP). The more pollen samples per time intervals and pollen records per grid cells, the more years within the 500 yrs time slice will be represented in the reconstruction. This implies that the number of years represented in a time-slice reconstruction varies in space and time.

A major limitation in applying REVEALS globally is requirement for information about the relative pollen productivity (RPP) of individual pollen taxa, which is currently largely lacking for the tropics. However, LandCover6k has been collecting RPPs for China, South-East India, Cameroon, Brazil and Argentina and pollen-based land-cover reconstructions will be available for at sufficient parts of the tropics to allow testing of the scenarios. Another limitation of REVEALS reconstructions is that RPP estimates are available for cultivated cereals but not for other cultivars or cropland weeds, so the LandCover6k reconstructions will generally underestimate cropland cover (Trondman et al., 2015). It may also be possible to use alternative pollen-based reconstructions of land cover changes, such as the Modern Analogue Approach (MAT: e.g. Tarasov et al., 2007; Zanon et al. 2018); pseudo-biomization (e.g. Fyfe et al., 2014) or STEPPS (Dawson et al., 2016). While none of these methods require RPPs, MAT and STEPPS can only be applied in regions where the pollen datasets have dense coverage (such as Europe and North America) and pseudo-biomization is affected by the non-linearity of the pollen-vegetation relationship that the REVEALS approach is designed to remove.

Comparison of the reconstructions of the extent of open land with the LULC deforestation scenarios will provide a first evaluation of the realism of the revised LULC scenarios (e.g. Kaplan et al., 2017). Underestimation or overestimation of open land in the LULC scenarios is not necessarily an indication that these scenarios are inaccurate because (a) pollen-based reconstructions cannot distinguish between anthropogenic and climatically determined natural open land (e.g. natural grasslands, steppes, wetlands) and (b) REVEALS underestimates cropland cover because there are no RPP estimates for cultivars other than cereals. However, overestimation of the area of open land in the LULC scenarios might suggest problems either in the archaeological inputs or their implementation, especially for times or regions when other evidence indicates cereals were the major crop. In this sense, despite potential problems, the LandCover6k pollen-based reconstructions of land cover will provide an important independent test of the revised LULC scenarios.

## 6. Testing the reliability of improved scenarios using climate-model simulations

A second test of the realism of the improved LULC scenarios is to examine whether incorporating LULC changes improves the realism of the simulated climate when compared to palaeoclimate reconstructions (Figure 8). The mid-Holocene (6000 years ago, 6 ka BP) is an ideal candidate for such a test because benchmark data sets of quantitative climate reconstructions are available (e.g. Bartlein et al., 2011), the interval has been a focus through multiple phases of PMIP and control simulations with no LULC have already been run, and evaluation of these simulations has identified regions where there are major discrepancies between simulated and reconstructed climates e.g. the observed expansion of northern hemisphere monsoons, climate changes over Europe, the magnitude of high-latitude warming, and wetter conditions in central Eurasia (Mauri et al., 2014; Harrison et al., 2015; Bartlein et al., 2017). There are discernible anthropogenic impacts on the landscape in many of these regions by 6 ka, although they are not as strong as during the later Holocene and they are not present everywhere. Nevertheless, the 6ka BP interval provides a good focus for testing improvements to the LULC scenarios. Such an evaluation would need to go beyond the global comparison made here (Figure 8) to regional comparisons to identify whether improvements in regions where there is a large anthropogenic impact on land cover do not result in a degradation in the simulated climate elsewhere.

## 7. Testing the reliability of improved scenarios using carbon-cycle models

Carbon-cycle modelling will be used as a further test of the realism of the improved LULC scenarios. Two constraints are available for testing the realism of past LULC reconstructions. First, reconstructions of LULC history must converge on the present-day state, which is relatively well constrained by satellite land-cover observations and national statistics on the amount of land under use. Reconstructing the extent of past LULC thus reduces to allocating a fixed total amount of land conversion from natural to agricultural use over time. More conversion in earlier periods implies less conversion in later periods. At the continental to global scale, cumulative LULC emissions scale linearly with the agricultural area. LULC scenarios that converge to the present-day state also converge to within a small range of cumulative historical emissions (Stocker et al., 2011; Stocker et al., 2017). Deviations from a linear relationship between extent and emissions are due to differences in biomass density in potential natural and agricultural vegetation of different regions affected by anthropogenic LULC. Differences in cumulative emissions for alternative LULC reconstructions with an identical present-day state are due to the long response time of soil carbon content following a change in carbon inputs and soil cultivation. Conserving the total extent of LULC (and allocating a fixed total expansion over time) is thus approximately equivalent to conserving cumulative historical LULC emissions. Thus, more LULC $CO_2$ emissions in earlier periods imply less $CO_2$ emissions in more recent periods.

The total C budget of the terrestrial biosphere provides a second constraint on LULC emissions through time. The net C balance of the land biosphere, which reflects the sum of all natural and anthropogenic effects on terrestrial C storage, can be reconstructed from ice-core data of past $CO_2$ concentrations and $\delta^{13}C$ composition (Elsig et al. 2009). Providing that all of the natural contributions to the land C inventory (e.g. the build up of natural peatlands: Loisel et al., 2014) can be specified from independent evidence, the anthropogenic sources can be estimated as the difference between the total terrestrial C budget and natural contributions (Figure 9) at any specific time.

Transient simulations with a model that simulates $CO_2$ emissions in response to anthropogenic LULC can be used to test the reliability of the LULC changes through time, by comparing results obtained

with prescribed LULC changes through time against a baseline simulation without imposed LULC.
This will necessitate making informed decisions about the fraction of land under cultivation that is
abandoned or left fallow each year, and the maximum extent of land affected by such episodic
cultivation. We envisage using several different offline carbon-cycle models for this purpose in order
to take account of uncertainties associated with inter-model differences. The carbon-cycle simulations
will be driven by climate outputs (temperature, precipitation and cloud cover) from an existing
transient climate simulation made with the ECHAM model (Fischer and Jungclaus, 2011) and $CO_2$
prescribed from ice-core records. The $CO_2$ emission estimates from these two simulations will then
be evaluated using C budget constraints. This evaluation will allow us to pinpoint potential
discrepancies between known terrestrial C balance changes and estimated LULC $CO_2$ emission in
given periods over the Holocene.
**8. Implementation of LULC in Earth System Model simulations**
We propose a series of simulations to examine the impact of LULC, using the revised LULC scenarios
from LandCover6k and building on experiments that are currently being run either in CMIP6-PMIP4
(*midHolocene, past1000*) or within PMIP although not formally included as CMIP6-PMIP4
experiments.
The *mid-Holocene* (and its corresponding *piControl*) is one of the PMIP entry cards in the CMIP6-
PMIP4 experiments (Kageyama et al., 2018; Otto-Bliesner et al., 2017) and it is therefore logical to
propose this period for LULC simulations. The LULC sensitivity experiment (*midHoloceneLULC*)
should therefore follow the CMIP6-PMIP4 protocol, that is it should be run with the same model
components and following the same protocols for implementing external forcings as used in the two
CMIP6-PMIP4 experiments (Table 1). Thus, if the *piControl* and *midHolocene* simulations is run
with interactive (dynamic) vegetation, then the *midHoloceneLULC* experiment should also be run
with dynamic vegetation in regions where there is no LULC change. For most models, this means
that the LULC forcing is imposed as a fraction of the grid cell and the remaining fraction of the grid
cell has simulated natural vegetation. These new mid-Holocene simulations would allow for a better
understanding of the relationship between climate changes and land-surface feedbacks (including
snow albedo feedbacks), and the role of water recycling at a regional scale. Thus, modelling groups
who are running the *midHolocene* experiment with a fully interactive carbon cycle could also run the
LULC experiment allowing atmospheric $CO_2$ to evolve interactively, subject to the simulated ocean
and land C balance.
The real strength of the revised LULC scenarios is to provide boundary conditions for transient
simulations. The CMIP6-PMIP4 simulation of 850-1850 CE (*past1000*) already incorporates LULC
changes as a forcing (Jungclaus et al. 2017), based on a harmonized data set that provides LULC
changes from 850 through to 2015 CE (Hurtt et al., 2017), which in turn draws on output from the
HYDE3.2 data set (Klein Goldewijk et al., 2017a). The *past1000* protocol (Jungclaus et al., 2017)
acknowledges that this default land-use data set is at the lower end of the spread in estimates of early
agricultural area indicated by other scenarios and recommends that modelling groups run additional
sensitivity experiments using alternative maximum and minimum scenarios. The revised scenarios
created by LandCover6k could be used as an alternative to these maximum and minimum scenarios.
Other than the substitution of the LandCover6k scenario, the specifications of other forcings would
then follow the recommendations for the CMIP6-PMIP4 *past1000* simulation.
A transient simulation for a longer period of the Holocene would provide a more stringent test of the
impact of LULC on the coupled earth system. We suggest that this transient simulation (*holotrans*)

should start from the pre-existing *midHolocene* simulation  to capitalise on the fact that the *midHolocene* simulation have been spun up for sufficiently long (Otto-Bleisner et al., 2017) to ensure that the ocean and land carbon cycle is in equilibrium at the start of the transient experiment (Table 2). In order to be consistent with the CMIP6-PMIP4 *midHolocene* protocol (Otto-Bleisner et al., 2017), changes in orbital forcing should be specified from Berger and Loutre (1991) and year-by-year changes in $CO_2$, $CH_4$ and $N_2O$ should be specified following Joos and Spahni (2008). LULC changes should be implemented by imposing crop and pasture area through time as specified in the revised LULC scenarios; elsewhere, the simulated vegetation should be active. It will be necessary to run the Holocene transient simulation in two steps. A first simulation (*holotrans  LULC*) should be run using prescribed atmospheric $CO_2$ concentration prescribed in the atmosphere even though the carbon cycle is fully interactive, because this will establish the consistency of the carbon cycle in the land surface model. However, once this is done it will be possible to re-run the simulations with interactive $CO_2$ emissions. Table 3 provides a summary of the proposed ESM simulations.

Unlike the situation for the mid-Holocene, where there is a global climate benchmark data set (Bartlein et al., 2011), quantitative evaluation of the *holotrans* simulated climate can only be made for key regions. Quantitative climate reconstructions through the Holocene are currently only available for Europe (Davis et al., 2003) and North America (Viau et al, 2006; Viau and Gajewski, 2009). However, there are time series reconstructions for individual sites outside these two regions (e.g. Nakagawa et al., 2002; Wilmshurst et al., 2007; Ortega-Rosas et al., 2008). Furthermore, the simulated time-course of $CO_2$ emissions can be compared to the ice core records.

The CMIP6-PMIP4 *mid-Holocene* simulations are stylized experiments, lacking several potential forcings (in addition to LULC), including changes in atmospheric dust loading, in solar irradiance, and volcanic forcing. We suggest that additional sensitivity tests could be run to take these additional forcings into account. In the case of solar and volcanic forcing, this would also ensure that the transient *holotrans* simulations mesh seamlessly with the *past1000* simulation. Changes in solar variability during the Holocene should be specified from Steinhilber et al. (2012). There are records of volcanic forcing for the past 2000 years (Sigl et al., 2015; Toohey and Sigl, 2017), and these are used in the *past1000* simulation. Observationally constrained estimates of the volcanic stratospheric aerosol for Holocene are currently under development (M. Sigl, pers comm.) and could be implemented as an additional sensitivity experiment when available. Changes in atmospheric dust loading are not included in the *past1000* simulation but are important during the earlier part of the Holocene (Pausata et al., 2016; Tierney et al., 2017; Messori et al., 2019). Although continuous reconstructions of dust loading through the Holocene are not available, it would be possible to use estimates for particular time-slices (Egerer et al., 2018) to test the sensitivity to this forcing.

**Outcomes and Perspectives**

LandCover6k has developed a scheme for using archaeological information to improve existing scenarios of LULC changes during the Holocene, specifically by using archaeological data to provide better estimates of regional population changes through time, better information on the date of initiation of agriculture in a region, more regionally specific information about the type of land use, and more nuanced information about land-use per capita than currently implemented in the LULC scenarios generated by HYDE and KK10. While the final global data set are still in production, fast-track priority products have been created and their impact on current scenarios is being tested.

Although the work of LandCover6k will provide more solid knowledge about anthropogenic modification of the landscape, some information will inevitably be missing and some key regions

will be poorly covered. There will still be large uncertainties associated with LULC scenarios. Documenting these uncertainties is an important goal of the LandCover6k project, and will allow the generation of multiple scenarios comparable to the "low-end", "high-end" scenarios used for e.g. in future projections. Furthermore, we have proposed a series of tests that will help to evaluate the realism of the final scenarios, based on independent evidence from pollen-based reconstructions of land cover, reconstructions of climate, and carbon-cycle constraints. These tests should help in identifying which of the potential LULC reconstructions are most realistic and constraining the sources of uncertainty.

We have proposed the use of offline vegetation-carbon-cycle simulations solely as a test of the realism of the revised LULC scenario. Quantifying the LULC contribution to $CO_2$ emissions during the Holocene would require additional simulations in which other forcings (climate, atmospheric $CO_2$, insolation) are kept constant. The difference in simulated total terrestrial C storage between these simulations and LULC simulations provides an estimate of *primary emissions* (Pongratz et al., 2014) and avoids additional model uncertainty regarding the sensitivity of land C storage to atmospheric $CO_2$ or climate being included in emission estimates. There are other sensitivity tests that would be useful. For example, vegetation-carbon-cycle models differ in their ability to account for gross land use transitions within grid cells (Arneth et al., 2017). This is critical for simulating effects of non-permanent agriculture where land is simultaneously abandoned and re-claimed within the extent of a model grid cell. Such shifting cultivation-type agriculture implies forest degradation in areas recovering from previous land use and leads to substantially higher LULC emissions compared to model estimates where only net land-use changes are accounted for (Shevliakova et al., 2009). It would therefore be interesting to run additional simulations accounting for net land use change, and indeed separating out the effects of wood harvesting and shifting cultivation.

We anticipate that it will be possible to incorporate realistic LULC for the mid-Holocene as part of the sensitivity experiments planned during PMIP4. Such experiments will complement the CMIP6-PMIP4 baseline experiments, by providing insights into whether discrepancies between simulated and observed 6 ka climate could be the result of incorrect specification of the land-surface boundary conditions. However, the incorporation of archaeological information into LULC scenarios clearly makes it possible to target other interesting periods for such experiments, for example to explore if land-use changes played a role in abrupt events such as the 4.2 ka event, or to examine the impact of population declines in the Americas as a consequence of European colonisation (1500-1750 CE) or the changes in land use globally during the Industrial era (post 1850 CE).

In addition to providing a protocol for the PMIP 6ka sensitivity experiments, we have devised a protocol for implementing the optimal LULC reconstructions for the Holocene in transient experiments. The goal here is to provide one of the necessary forc- ings that could be used for transient simulations in future phases of PMIP. This will allow an assessment of LULC in these simulations, and therefore help address is- sues that are a focus for other MIPs e.g. LUMIP or LS3MIP. When these new forcings are created, they will be made available through the PMIP4 website (https: //pmip4.lsce.ipsl.fr/doku.php/exp_design:lgm, PMIP4 repository, 2017) and the ESGF Input4MIPS repository (https://esgf-node.llnl.gov/projects/input4mips/, with details pro- vided in the "input4MIPs summary" link). Modelling groups who run either equilibrium or transient experiments following this protocol are encouraged to follow the standard CMIP protocol of archiving their simulations through the ESFG.

**Acknowledgements.** LandCover 6k is a working group of the Past Global Changes (PAGES)

programme, which in turn received support from the Swiss Academy of Sciences. We thank PAGES for their support for this activity. The land use group also received funding under the Holocene Global Landuse International Focus Group of INQUA. SPH acknowledges funding from the European Research Council for "GC2.0: Unlocking the past for a clearer future". MJG thanks the Swedish Strategic Research Area MERGE (Modelling the Regional and Global Earth System Model) and Linnaeus University's faculty of Health and Life Sciences (Kalmar, Sweden) for financial support. We thank Joy Singarayer for providing the climate model outputs that were used to generate Figure 8 and Guangqi Li for assistance in producing this figure. B.D.S. was funded by ERC H2020-MSCA-IF-2015, grant number 701329. The dataset for Figure 5 was generated from the 'Cultivating Societies: Assessing the Evidence for Agriculture in Neolithic Ireland' project, supported by the Heritage Council, Ireland under the INSTAR programme 2008–2010 (Reference 16682 to Whitehouse, Schulting, Bogaard and McClatchie).

**Author Contributions.** SPH, MJG, BDS, MVL, KKG wrote the first draft. SPH, PB, FSRP contributed to the design of the climate model experiments, BS and TK to the design of the carbon-cycle simulations. The figures were contributed by JK (Fig. 1), BS (Fig. 2, Fig 9.), MVL (Fig. 3), OB (Fig. 4), NJW (Fig. 5), KKG (Fig. 6), AD (Fig. 7), SPH (Fig. 8). All authors contributed to the final version of the paper.

**Competing Interests.** No competing interests.

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

**Figure and Table Captions**

Figure 1: Land use at ca 6000 years ago (6ka BP, 4000 years BCE)  from the two widely used global historical land-use scenarios HYDE 3.2 (top panel, Klein Goldewijk et al. 2017a) and KK10 (bottom panel, Kaplan et al. 2011), illustrating the large disagreement between LULC scenarios at a regional scale. In both scenarios, the land-sea mask and lake areas are for the present day.

Figure 2: Proposed scheme for developing robust LULC scenarios through iterative testing and refinement, as input to Earth System Model (ESM) simulations. The archaeological inputs developed in Phase 1 can be used independently or together to improve the LULC reconstructions; iterative testing of the LULC scenario reconstruction (Phase 2) will ensure that these inputs are reliable before they are used of ESM simulations (Phase 3). The uppermost three LULC simulations capitalize on already planned baseline simulations without LULC; the lowermost two simulations are envisaged as new sensitivity experiments.

Figure 3: Reconstruction of changes in population size in the Iberian Peninsula during the Holocene (9000 to 2000 BP, 9ka to 2ka BP) using summed probability distributions (SPDs) of radiocarbon dates (data after Balsera et al., 2015). The red line indicates the onset of agriculture in the region. The lower panels show areas under human use at 6ka (left) and 4ka (right) using kernel density estimates, where the white dots are actual archaeological sites and the shading shows the implied density of occupation.

Figure 4: The hierarchical scheme of land-use classes used for global mapping in LandCover6k (updated from Morrison et al, 2018).

Figure 5: An example of regional land-use mapping. The upper panels show the distribution of known archaeological sites superimposed on kernel density estimates of the extent of land-use based on the density of observations, and the lower panels show these data superimposed on the LandCover6k land-use classes for the Middle Neolithic (3600-3400 years BCE, 5600-5400 years BP, 5.6-5.4 ka BP) (left panels) and the Early Neolithic (3750-3600 years BCE, 5750-5600 years BP, 5.7-5.6 ka BP) (right panels) of Ireland. Data points derive from [14]C dated archaeological sites and distributions of settlements and monuments that have been assigned to each archaeological period following the dataset published in McLaughlin et al. (2016). The assigned land-use classes are inferred from archaeological material from one (or more) sites within the grid box. It should not be assumed that the whole gridcell was being used for agriculture during the Middle and Early Neolithic. Informed assessment suggests that agricultural land (crop growing and grazing, combined) probably occupied between 10-15% of the total grid area in the low-level food production regions of the eastern and western coastal areas, whilst agricultural land likely represents 5% or less of the total grid cell area in inland areas.

Figure 6: Schematic illustration of the proposed implementation of [14]C-based population estimates, date of first agriculture, land-use maps, and land-use per capita information in the HYDE model (here indicated as HYDE3.x). The archaeological data are represented as values for a grid cell in geographic space at a given time for date of first agriculture and land use, but as a time series for a specific grid cell for population and land-use per capita. In the case of population estimates, date of first agriculture and land-use per capita data, we show the initial estimate and the revised estimate after taking the archaeological information into account in the HYDE3.x plot. It should be assumed in the case of the land-use mapping that the original estimate was that there was no land use in this region.

Figure 7: Northern extratropical (>40°N) mean fractional cover of open land at 200 years ago (0.2ka

BP) and 6000 years ago (6ka BP estimated using REVEALS, and the difference in fractional cover
between the two periods (0.2 ka BP - 6ka BP), where red indicates an increase in open land and blue
a decrease (after Dawson et al., 2018).
Figure 8: Quantitative comparison of the change in climate between the mid-Holocene (6ka) and the
pre-industrial period as shown by pollen-based reconstructions gridded to 2 x 2° resolution to be
compatible with the model resolution (from Bartlein et al., 2011) and in simulations with and without
the incorporation of land-use change (from Smith et al., 2016). This figure illustrates the approach
that will be taken to evaluate the impact of new LULC scenarios on climate. The imposed land-use
changes at 6000 years ago (6ka BP) were derived from the KK10 scenario (Kaplan et al., 2011). The
plots show comparisons of mean annual temperature (MAT), mean temperature of the coldest month
(MTCO) and mean annual precipitation (MAP) for the northern extratropics (north of 30° N), where
each dot represents a model grid cell where comparisons with the pollen-based reconstructions is
possible. Although the incorporation of land use produces somewhat warmer and wetter climates in
these simulations, overall the incorporation of land-use produces no improvement of the simulated
climates at sites with pollen-based reconstructions.
Figure 9: Illustration of the terrestrial C budget approach to evaluate LULC. The total terrestrial C
balance (green circle 'total') is constrained by ice core records of $CO_2$ and its isotopic signature ($\delta^{13}$C).
Estimates for C balance changes of different natural land carbon cycle components (e.g., peatlands,
permafrost, forest expansion/retreat, desert greening) can are estimated independently (blue slices
'Natural components') either from empirical upscaling of site-scale observations or from model-based
analyses (BGC models forced with varying climate).The remainder (yellow slice 'remainder') is then
calculated as the total terrestrial C balance (green circle 'total') minus the sum of separate estimates
of natural components (blue slices 'Natural components') The remainder is effectively the emissions
resulting from LULC changes, and can therefore be compared to LULC $CO_2$ emission estimates by
carbon-cycle models.
Table 1. Boundary conditions for CMIP6-PMIP4 and the mid-Holocene LULC experiments. The
boundary conditions for the CMIP6-PMIP4 piControl and midHolocene are described in Otto-
Bleisner et al. (2017) and are given here for completeness.
Table 2. Boundary conditions for baseline PMIP Holocene transient (6 ka BP to 1850 CE) and LULC
transient simulations
Table 3. Summary of proposed simulations.

Figure 1: Land use at ca 6000 years ago (6ka BP, 4000 years BCE) from the two widely used global
historical land-use scenarios HYDE 3.2 (top panel, Klein Goldewijk et al. 2017a) and KK10 (bottom
panel, Kaplan et al. 2011), illustrating the large disagreement between LULC scenarios at a regional
scale. In both scenarios, the land-sea mask and lake areas are for the present day.

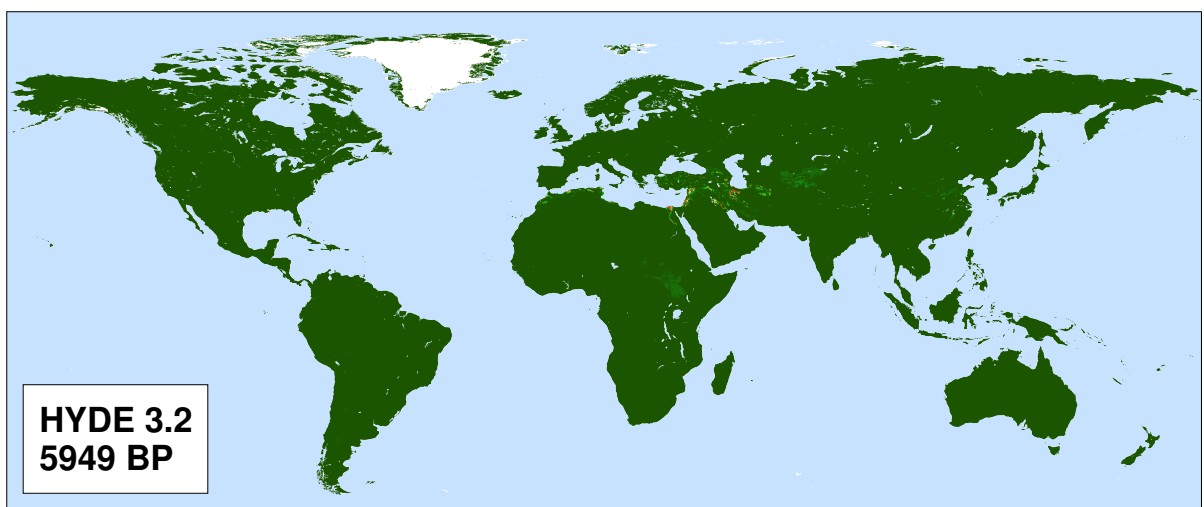

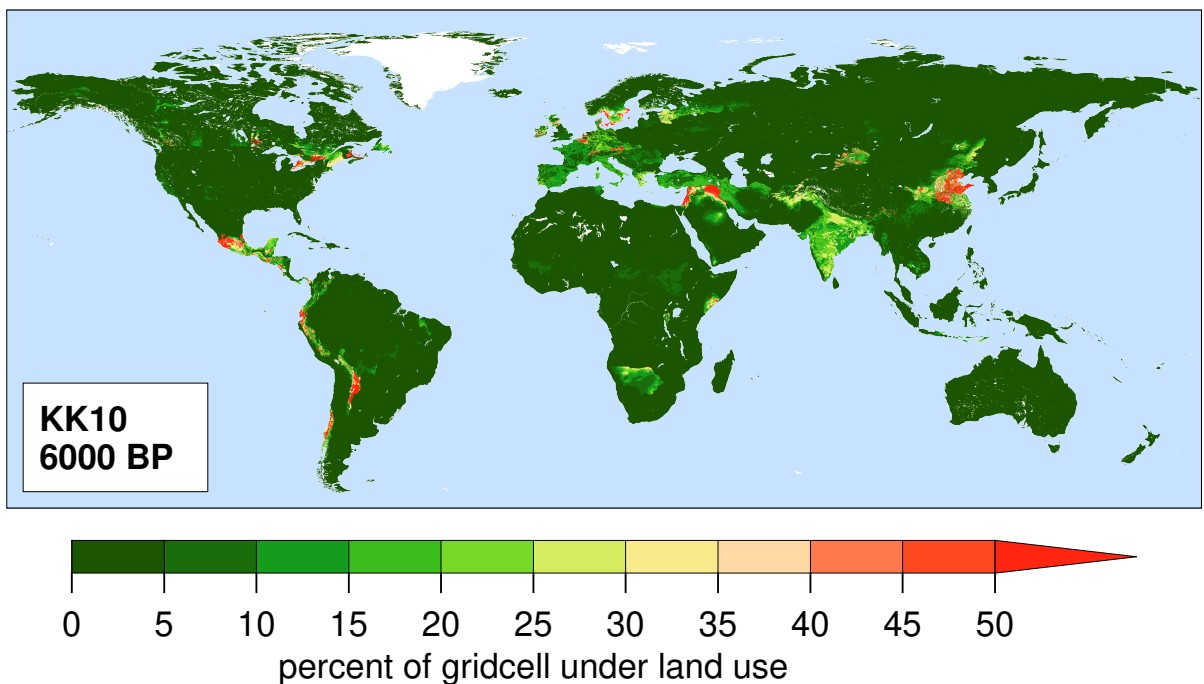


Figure 2: Proposed scheme for developing robust LULC scenarios through iterative testing and refinement, as input to Earth System Model (ESM) simulations. The archaeological inputs developed in Phase 1 can be used independently or together to improve the LULC reconstructions; iterative testing of the LULC scenario reconstruction (Phase 2) will ensure that these inputs are reliable before they are used of ESM simulations (Phase 3). The uppermost three LULC simulations capitalize on already planned baseline simulations without LULC; the lowermost two simulations are envisaged as new sensitivity experiments.

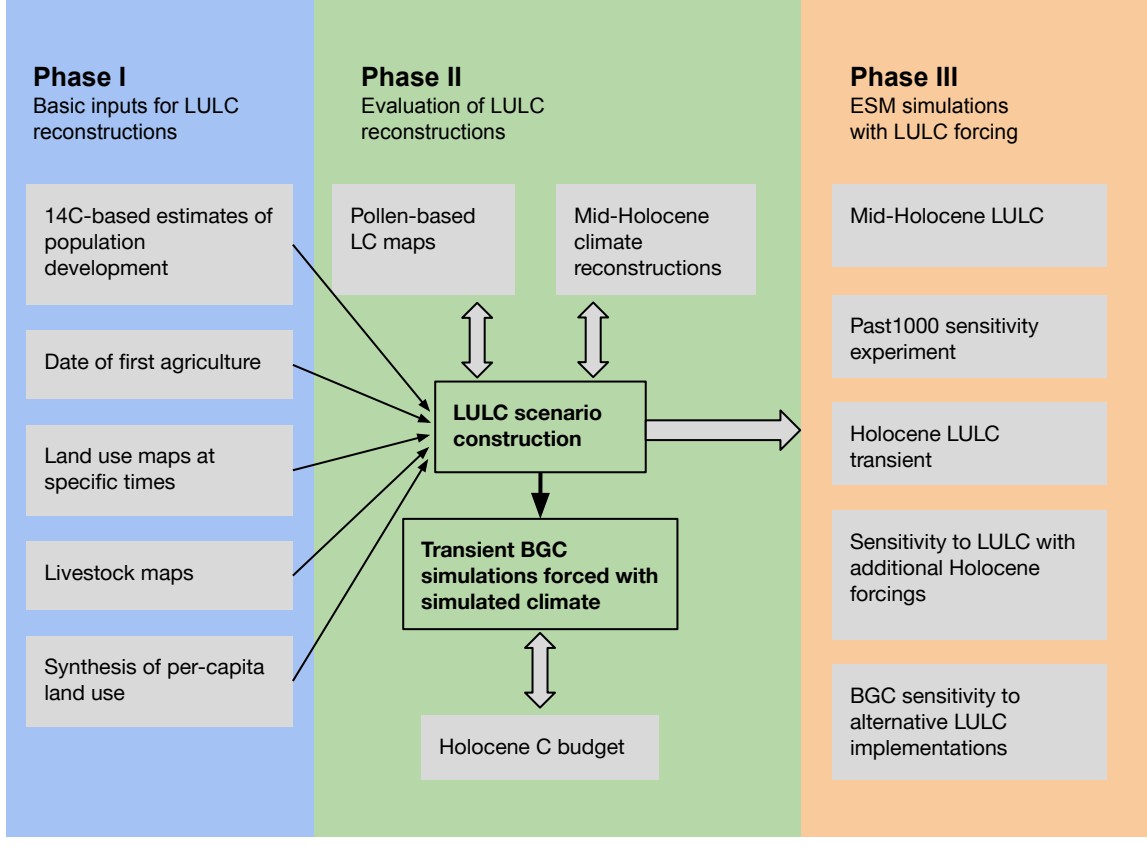

Figure 3: Reconstruction of changes in population size in the Iberian Peninsula during the Holocene
(9000 to 2000 BP, 9ka to 2ka BP) using summed probability distributions (SPDs) of radiocarbon
dates (data after Balsera et al., 2015). The red line indicates the onset of agriculture in the region. The
lower panels show areas under human use at 6ka (left) and 4ka (right) using kernel density estimates,
where the white dots are actual archaeological sites and the shading shows the implied density of
occupation.

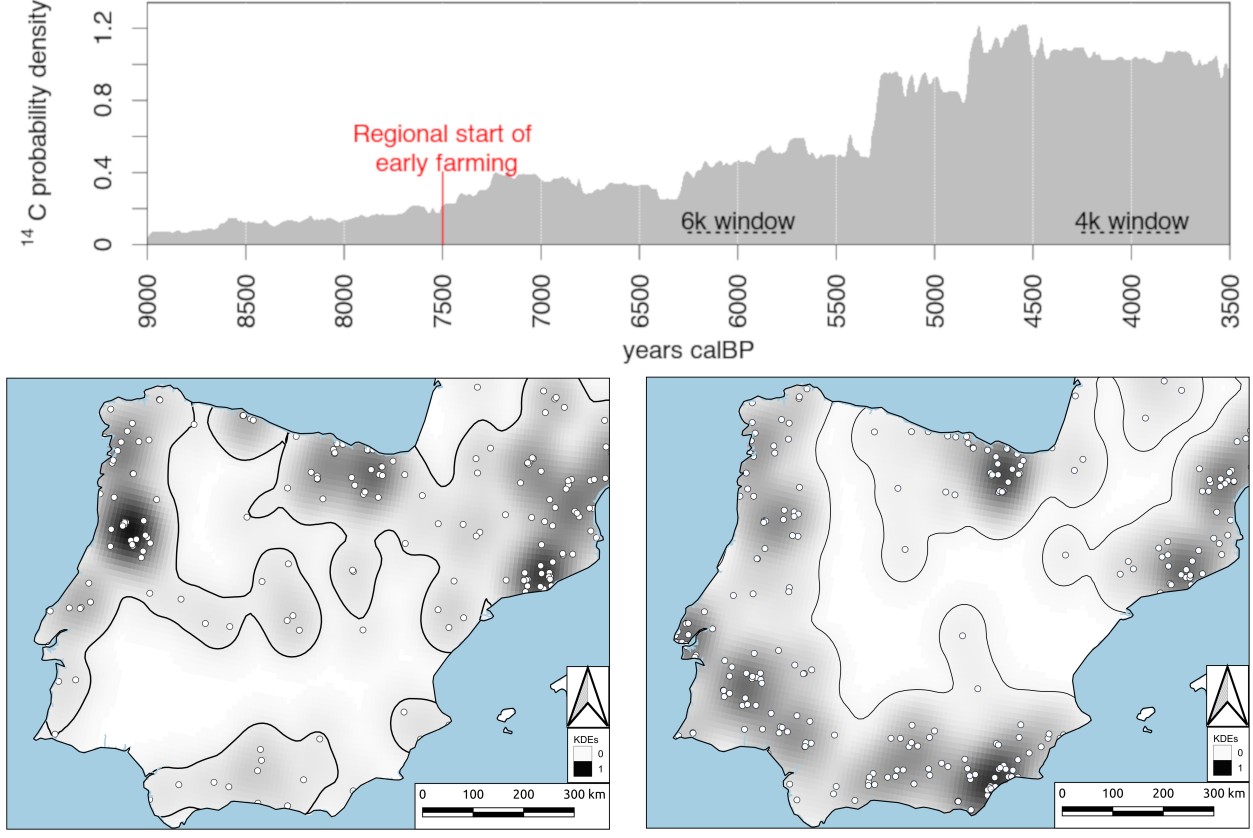


Figure 4: The hierarchical scheme of land-use classes used for global mapping in LandCover6k
(updated from Morrison et al, 2018).

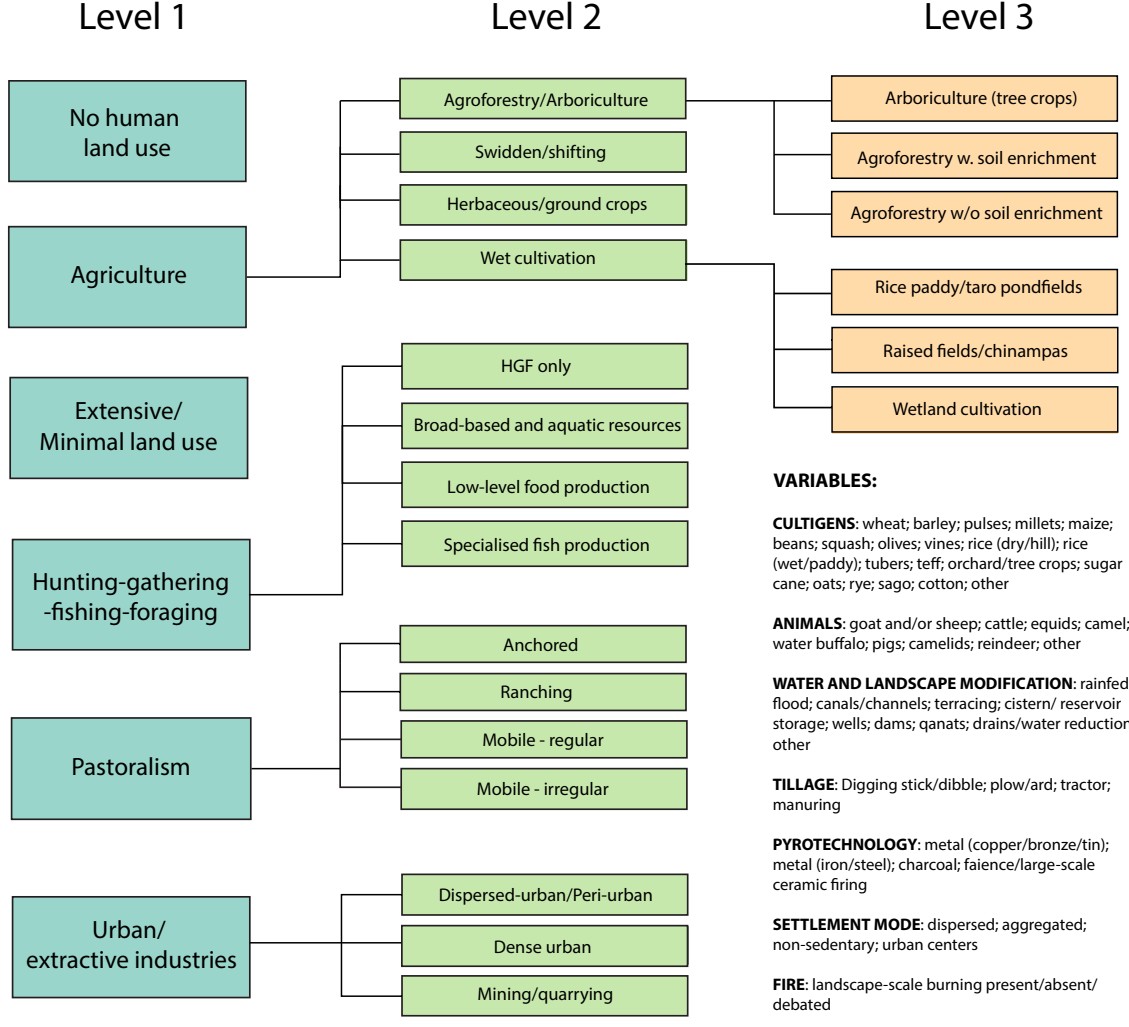


Figure 5: An example of regional land-use mapping. The upper panels show the distribution of known archaeological sites superimposed on kernel density estimates of the extent of land-use based on the density of observations, and the lower panels show these data superimposed on the LandCover6k land-use classes for the Middle Neolithic (3600-3400 years BCE, 5600-5400 years BP, 5.6-5.4 ka BP) (left panels) and the Early Neolithic (3750-3600 years BCE, 5750-5600 years BP, 5.7-5.6 ka BP) (right panels) of Ireland. Data points derive from [14]C dated archaeological sites and distributions of settlements and monuments that have been assigned to each archaeological period following the dataset published in McLaughlin et al. (2016). The assigned land-use classes are inferred from archaeological material from one (or more) sites within the grid box. It should not be assumed that the whole gridcell was being used for agriculture during the Middle and Early Neolithic. Informed assessment suggests that agricultural land (crop growing and grazing, combined) probably occupied between 10-15% of the total grid area in the low-level food production regions of the eastern and western coastal areas, whilst agricultural land likely represents 5% or less of the total grid cell area in inland areas.

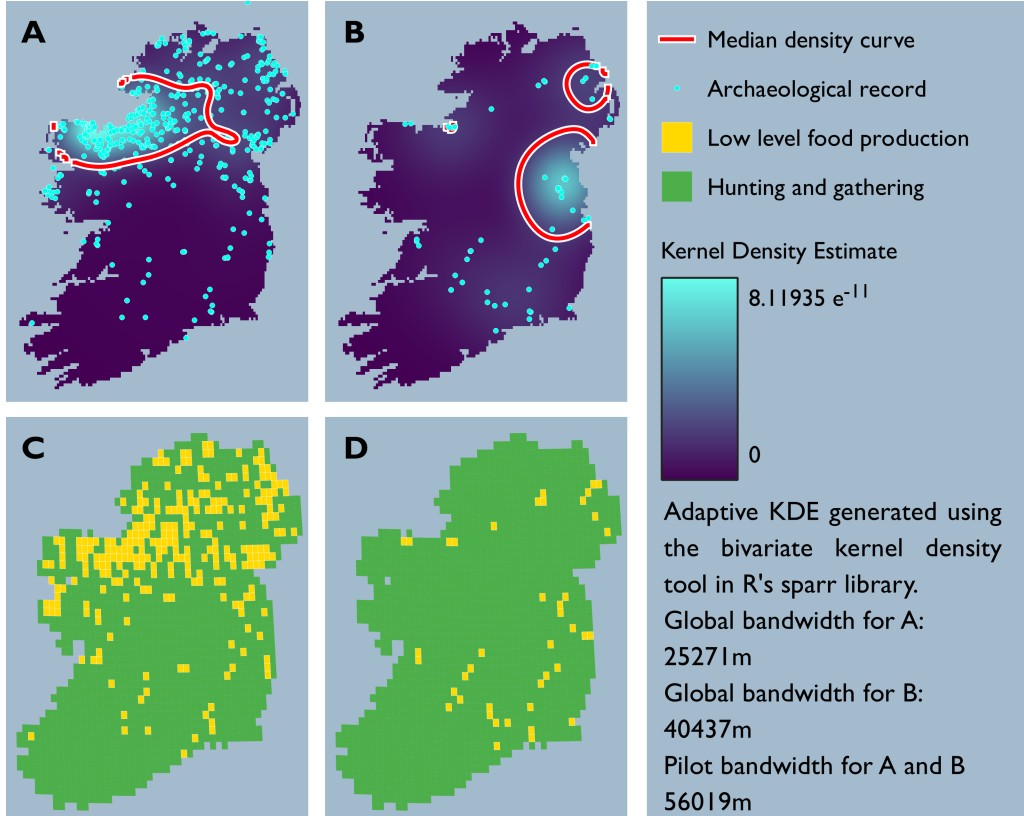

Figure 6: Schematic illustration of the proposed implementation of $^{14}$C-based population estimates,
date of first agriculture, land-use maps, and land-use per capita information in the HYDE model (here
indicated as HYDE3.x). The archaeological data are represented as values for a grid cell in geographic
space at a given time for date of first agriculture and land use, but as a time series for a specific grid
cell for population and land-use per capita. In the case of population estimates, date of first agriculture
and land-use per capita data, we show the initial estimate and the revised estimate after taking the
archaeological information into account in the HYDE3.x plot. It should be assumed in the case of the
land-use mapping that the original estimate was that there was no land use in this region.

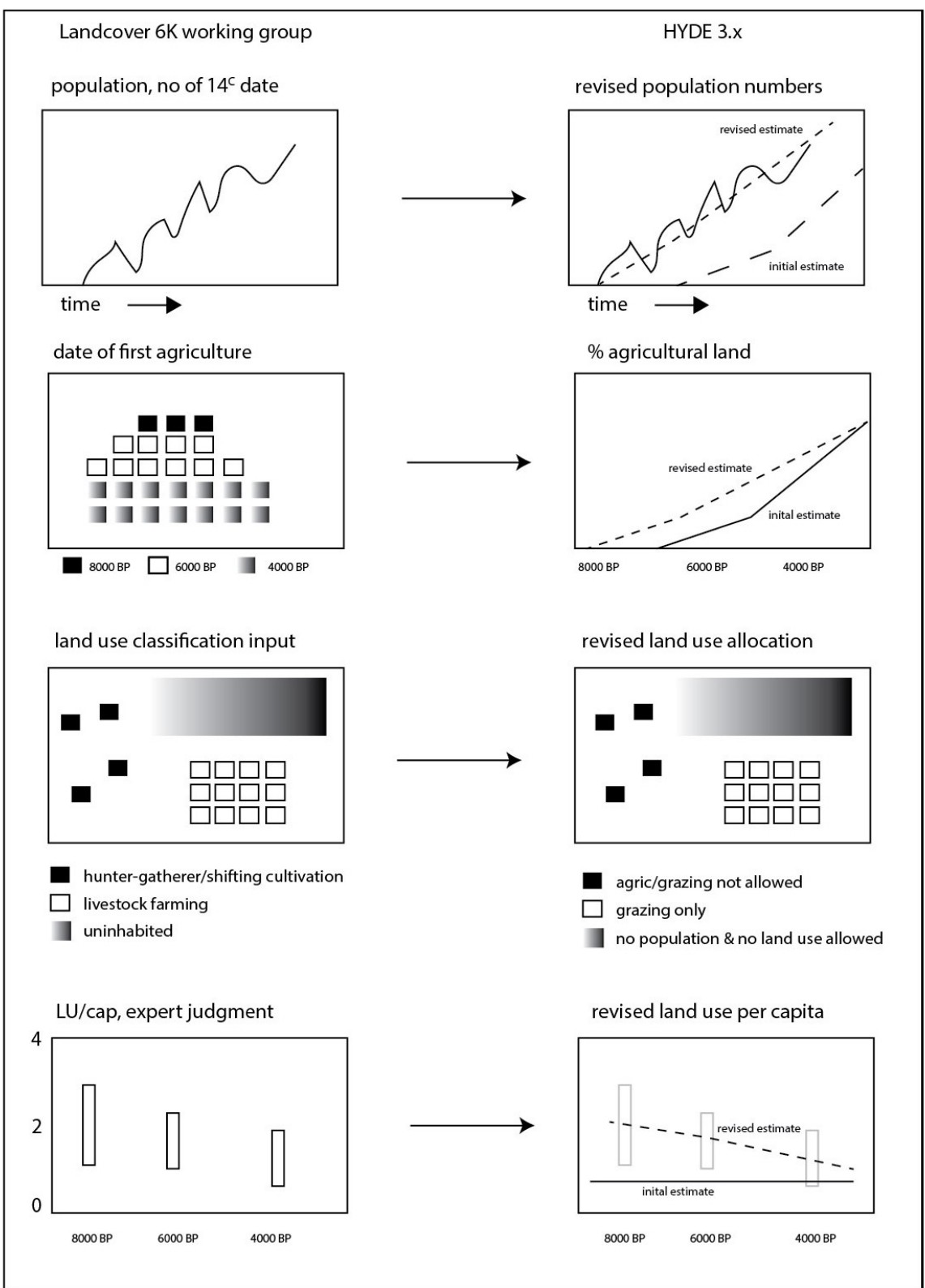

Figure 7: Northern extratropical (>40°N) mean fractional cover of open land at 200 years ago (0.2ka BP) and 6000 years ago (6ka BP estimated using REVEALS, and the difference in fractional cover between the two periods (0.2 ka BP - 6ka BP), where red indicates an increase in open land and blue a decrease (after Dawson et al., 2018).

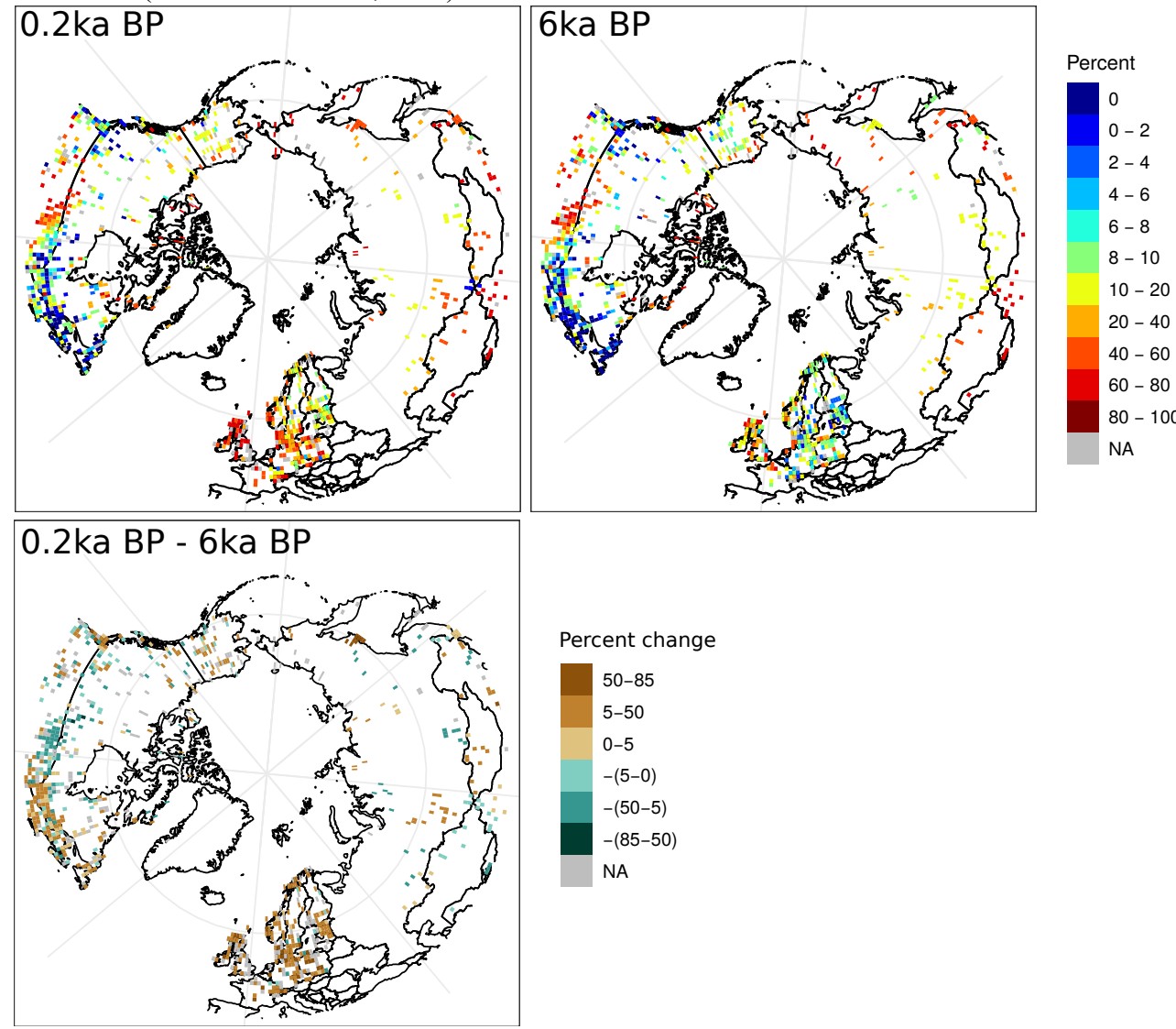

Figure 8: Quantitative comparison of the change in climate between the mid-Holocene (6ka) and the pre-industrial period as shown by pollen-based reconstructions gridded to 2 x 2° resolution to be compatible with the model resolution (from Bartlein et al., 2011) and in simulations with and without the incorporation of land-use change (from Smith et al., 2016). This figure illustrates the approach that will be taken to evaluate the impact of new LULC scenarios on climate. The imposed land-use changes at 6000 years ago (6ka BP) were derived from the KK10 scenario (Kaplan et al., 2011). The plots show comparisons of mean annual temperature (MAT), mean temperature of the coldest month (MTCO) and mean annual precipitation (MAP) for the northern extratropics (north of 30° N), where each dot represents a model grid cell where comparisons with the pollen-based reconstructions is possible. Although the incorporation of land use produces somewhat warmer and wetter climates in these simulations, overall the incorporation of land-use produces no improvement of the simulated climates at sites with pollen-based reconstructions.

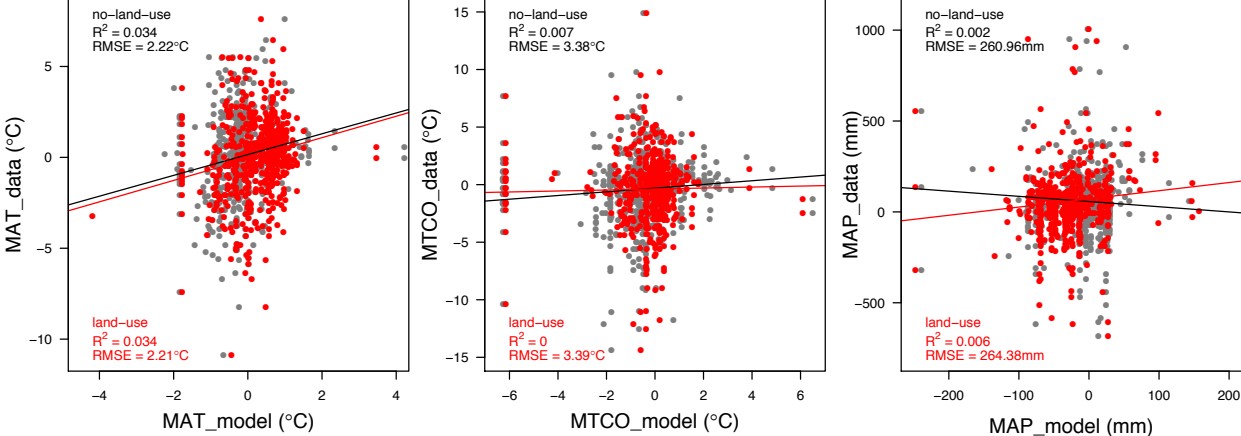

Figure 9: Illustration of the terrestrial C budget approach to evaluate LULC. The total terrestrial C
balance (green circle 'total') is constrained by ice core records of $CO_2$ and its isotopic signature ($\delta^{13}C$).
Estimates for C balance changes of different natural land carbon cycle components (e.g., peatlands,
permafrost, forest expansion/retreat, desert greening) can are estimated independently (blue slices
'Natural components') either from empirical upscaling of site-scale observations or from model-based
analyses (BGC models forced with varying climate).The remainder (yellow slice 'remainder') is then
calculated as the total terrestrial C balance (green circle 'total') minus the sum of separate estimates
of natural components (blue slices 'Natural components') The remainder is effectively the emissions
resulting from LULC changes, and can therefore be compared to LULC $CO_2$ emission estimates by
carbon-cycle models.

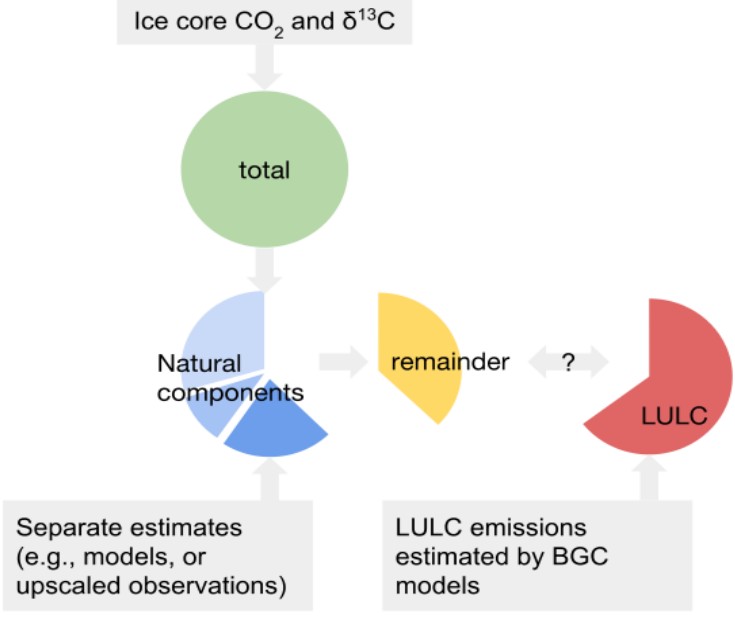



**Table 1**. Boundary conditions for CMIP6-PMIP4 and the mid-Holocene LULC experiments. The
boundary conditions for the CMIP6-PMIP4 *piControl* and *midHolocene* are described in Otto-
Bleisner et al. (2017) and are given here for completeness.

| Boundary conditions | | 1850CE (DECK *piControl*) | 6ka (*midHolocene*) | 6ka LULC (*midHoloceneLULC*) |
|---|---|---|---|---|
| Orbital parameters | Eccentricity | 0.016764 | 0.018682 | 0.018682 |
| | Obliquity | 23.459 | 24.105 | 24.105 |
| | Perihelion – 180 | 100.33 | 0.87 | 0.87 |
| | Vernal equinox | Noon, 21 March | Noon, 21 March | Noon, 21 March |
| Greenhouse gases | Carbon dioxide (ppm) | 284.3 | 264.4 | 264.4 |
| | Methane (ppb) | 808.2 | 597.0 | 597.0 |
| | Nitrous oxide (ppb) | 273.0 | 262.0 | 262.0 |
| | Other GHG | DECK *piControl* | 0 | 0 |
| Other boundary conditions | Solar constant | TSI: 1360.747 | As *piControl* | As *piControl* |
| | Palaeogeography | Modern | As *piControl* | As *piControl* |
| | Ice sheets | Modern | As *piControl* | As *piControl* |
| | Vegetation | Interactive | Interactive | pasture and crop distribution prescribed from the revised scenario |
| | | DECK *piControl* | As *piControl* | pasture and crop distribution prescribed from the revised scenario |
| | Aerosols | interactive | Interactive | Interactive |
| | | DECK *piControl* | As *piControl* | As *piControl* |


**Table 2**. Boundary conditions for baseline PMIP Holocene transient (6 ka BP to 1850 CE) and LULC
transient simulations

| | | Mode | Source/Value | LULC experiment |
|---|---|---|---|---|
| Orbital parameters | | transient | | As baseline simulation |
| Greenhouse gases | $CO_2$ | transient | Dome C | As baseline simulation |
| | $CH_4$ | | Combined EPICA & GISP record | As baseline simulation |
| | $N_2O$ | | Combined EPICA NGRIP, & TALDICE record | As baseline simulation |
| Solar forcing | | transient | Steinhilber et al. (2012) | As baseline simulation |
| Volcanic forcing | | transient | To be determined | As baseline simulation |
| Palaeogeography | | Constant at PI values | Modern | As baseline simulation |
| Ice sheets | | Constant at PI values | Modern | As baseline simulation |
| Vegetation | | interactive | | LC6k transient pasture and crop distribution imposed |
| Aerosols | | Constant at PI values | | As baseline simulation |


**Table 3**: Summary of proposed simulations.

| Name | Mode | Purpose |
|---|---|---|
| *piControl* | equilibrium | Standard CMIP6-PMIP4 simulation |
| *midHolocene* | equilibrium | Standard CMIP6-PMIP4 simulation |
| *midHoloceneLULC* | equilibrium | Sensitivity to LULC changes |
| *holotrans* | transient | Baseline fully transient simulation from 6ka onwards, with no LULC |
| *holotrans_LULC* | transient | Fully transient simulation from 6ka onwards, with LULC imposed |
