# Peer review of "Development and testing of scenarios for implementing land use and land cover changes during the Holocene in Earth System Model Experiments"

_Geoscientific Model Development, 2019_

## Short Comment (SC1) · 25 Oct 2019

Thank you for your submission to GMD. Currently the data availability section of the paper does not appear to be compliant with GMD policy. The manuscript needs to be fully compliant before a revised manuscript can be accepted for publication in GMD.

The issues are as follows:

1. "climate model forcing data sets, their references, and their code will be provided on the PMIP4 website". GMD does not allow embargoed data. The data should all have been publicly archived before the manuscript was submitted. It must

certainly be publicly archived before a revised manuscript could be accepted for publication in GMD.

2. The PMIP4 website does not appear to comply with the standards required for archiving of data presented in GMD papers. Project websites are not usually acceptable archive locations as they lack the persistence, non-revocability, and persistent identifiers required. Please see the GMD code and data policy. If the PMIP4 website does satisfy those requirements, then evidence for this needs to be presented, and the data should be referenced by persistent identifier (e.g. DOI) rather than URL. Otherwise, the data needs to be archived on a suitable location, such as Pangaea.

3. The reference to Pangea is confusing. I think you are saying that the outputs of the experiments resulting from this MIP must be archived to Pangaea. That is completely reasonable, but then that should form part of the protocol. This section of the manuscript is for documenting where the data that this paper depends on can be found. In this case, this is the input data for the models.

For further details on GMD code and data archiving requirements, please see the recent editorial

---

## Author Comment (AC1) · 29 Oct 2019

We apologise if we have caused confusion through our inclusion of a data availability section in the manuscript. This manuscript describes the protocol for a set of envisaged simulations. As such, there are no data accompanying the paper.

The climate model forcings that say should be used for the planned LULC simulations are standard PMIP data sets. They have been referenced in other PMIP protocol papers (e.g. Otto-Bleisner et al., GMD 10 (2017); Kageyama et al. GMD 10 (2017) as follows:

[Figure]

All the forcing data sets, their references, and their code can be found on the PMIP4 website (https: //pmip4.lsce.ipsl.fr/doku.php/exp_design:lgm, PMIP4 repository, 2017). The forcings will also be added to the ESGF Input4MIPS repository (https://esgf-node.llnl.gov/projects/input4mips/, with details provided in the "input4MIPs summary" link).

We followed this format in our Code section.

Model outputs that will be generated by groups following the protocol would normally be archived in the ESFG archive, following international standards and practice for CMIP6

The reference to PANGAEA was designed to indicate where validation data sets would be archived once they have been produced. Again, this paper indicates the type of validation that is envisaged as part of the protocol but does not present these data sets per se.

We will remove the Data Availability section from the manuscript, since this paper only describes the protocol for running experiments. In this case, we would modify the final paragraph of the paper to indicate that groups wishing to run these simulations should follow standard CMIP practice for archiving as follows:

In addition to providing a protocol for the PMIP 6ka sensitivity experiments, we have devised a protocol for implementing the optimal LULC reconstructions for the Holocene in transient experiments. The goal here is to provide one of the necessary forcings that could be used for transient simulations in future phases of PMIP. This will allow an assessment of LULC in these simulations, and therefore help address issues that are a focus for other MIPs e.g. LUMIP or LS3MIP. When these new forcings are created, they will be made available through the PMIP4 website (https: //pmip4.lsce.ipsl.fr/doku.php/exp_design:lgm, PMIP4 repository, 2017) and the ESGF Input4MIPS repository (https://esgf-node.llnl.gov/projects/input4mips/, with details provided in the "input4MIPs summary" link). Modelling groups who run either equilibrium or transient experiments following this protocol are encouraged to follow the standard

CMIP of archiving their simulations through the ESFG.

---

## Short Comment (SC2) · 14 Nov 2019

**Almut Arneth**

almut.arneth@kit.edu

Received and published: 14 November 2019

This is a well written paper that describes possible ways forward to improve historical land use change estimates, globally. Improving reconstructions of land use change is critical, given its impact on past vegetation cover, climate change, carbon cycle. And while the observational data-sets to undertake this endeavour necessarily are limited it is nonetheless a crucial starting point.

Some of my questions below arise most likely from being interested in land-use change but I'm not a paleo-expert. Most importantly, some paragraphs/(sub)sections of the paper could gain by providing a little bit more detail 'how' exactly the new approach will

be put in practise and how exactly the improvement might be envisaged.

Before I get into these there is one major aspect that seems missing from the approach. People need not only to eat, they also need to cook and heat, and to live. Has the group not discussed to -in addition to archaeological data- to also mine written historical records? This is probably most relevant for the last 1000+ years (rather than mid-Holocene), but surely there can be assumptions about wood requirements for building materials (analogue to a per-capita area needed to be fed: how many people would live in an 'average' house/farm and how much would this would need), shipping fleets (records from shipyards), charcoal making, furnaces for metal forging etc. I would imagine that at least in some regions this would have contributed perhaps already many centuries ago to deforestation. Could the authors comment on this aspect? To me this seems an obvious next step.

Lines 63-65: For correctness, I would avoid using the term "feedback" here in the sense of change in process A affects process B, feeding back to A. LUC impacts on the carbon cycle are nothing more than an additional emission (or uptake), similar to other anthropogenic emissions, and the biophysical processes are related to albedo or ET change – but these are not feedbacks.

Lines 89-99: might be worth pointing out that the large discrepancies between Hyde and KK10 arise mostly from the assumptions about per-capital land requirements; to my knowledge their estimates of historical population changes through time (at least global totals) are more or less the same.

Lines 125-132: Given that these MIPs are already well under way, could you pls comment how realistic it is that the communities will be able to take up these protocols in time? Is it not more likely that the work will be most useful for many other studies that may not follow the tight schedule of the current AR6 MIP-frenzy, including work that would be useful also in context of the IPBES; and/or might feed into the next IPCC cycle ? Lines 141/142: style; one 'required/requirements' might be sufficient...?
Figure 4: 'Wetland cultivation' in Level 3 – would that mean wetland drainage for agriculture? I assume it does, please clarify.

Lines 146-162 – bit of an unspecific list, can be more precise, give more concrete examples?

Section 3.1 – this section wasn't entirely clear to me. What samples are we talking about exactly, what is being dated, where do the samples come from? Could you provide an illustrative example?

Figure 5: I liked the Figure, is nice to see a concrete, illustrative example of the planned approach. However, it was not entirely obvious to me what the top and bottom panels in Fig. 5 are meant to convey: is it to show the improvements that can be made by adding the new information to the existing LandCover 6a? Or what is exactly the added value of the two combined? And what's the reasoning behind the 10-15% and the 5% mentioned in lines 269/270?

Lines 288/289: how do you obtain information about past irrigation? From archaeological data (irrigation structures?) I assume? Likewise, per-capita land needs surely change over time, agreed. But how can these estimates be obtained, could you provide more explanation and/or references to methods as to how to do this?

Figure 6, just for illustrative purpose only: the panels 'land use classification input' and 'revised land use allocation' look identical, might be illustrative to not only change the legend but also the drawing. Line 327-329: what's the basis for the optimism that 'eventually' these pollen-based reconstructions will also be available elsewhere (presumably: the tropics), is there initial work that points in that direction? And what's the pros/cons of the "other" pollen-based reconstructions that are mentioned?

Lines 385/386: "known" today is not quite true unfortunately. There are still sizeable discrepancies in today's land cover estimates in terms of major classes such as cropland, pasture, forest, 'other' (let alone in the degree to which these are being used).
Partially this arises from disagreements in terms of how a pasture or forest is defined. There is no need to add a long discussion but pls. revise the sentence slightly to express that there is also uncertainty for today.

Lines 383-399: The scaling aspect is important. However, cumulative LUC C emissions differ substantially depending on whether "net" or "gross" area changes are being calculated. The total agricultural area might be the same in both approaches, but the 'gross' approach considers expansion and reduction that might occur within a gridcell. The most prominent example is shifting cultivation, and today is mostly restricted to tropical regions. However, others have pointed out that such gross transition of course also are relevant on other parts of the world (see e.g., Fuchs et al., GCB, 2015), and were possibly even more so further back in time. The challenges that arise from this aspect are mentioned later in the Outcomes section but I wonder if it's not better to introduce these already here.

**GMDD**

---

## Short Comment (SC3) · 18 Nov 2019

**Comments on "Development and testing of scenarios for implementing Holocene LULC in Earth System Model Experiments" by Sandy Harrison et al.**

This is an interesting manuscript that deals with formulating and describing a highly needed protocol for constructing and improving land use and land cover (LULC) changes over the Holocene with the ambition to derive LULC scenarios for use in climate modeling applications. Methods of incorporating archeological data for reconstruction of LULC changes are described together with methods of how to evaluate the results and potentially improving them. Such an evaluation could involve climate model experiments and/or carbon cycle model experiments. Deviations between models when forced with reconstructed LULC and the reconstructed climate could then be used for pointing at regions where reconstructions need more work.

I'm not an expert in land-use or past changes in land-use but as a climate modeler with some limited experience in paleoclimate modelling I think that the paper would benefit from some more detailed discussion of potential limitations with the formulated strategy. In particular, the results illustrating the methods show: large spread, poor correlation and small differences between experiments with and without land-use (Figure 8). This could compromise the idea constraining land use change by climate model simulations. In parallel to the climate model uncertainty, what is the uncertainty associated with the carbon cycle models proposed to be used for constraining the land use? Is it small enough to allow for a meaningful estimate of land use? I think the paper would benefit from a more in-depth discussion about these uncertainties. Consideration could also be given if there would be a place for more detailed regional and local studies to further constrain land use?

General comments:

Some words and concepts are quite difficult for a climate modeler (definition of time periods like the Holocene and Mesolithic and Neolithic times, taphonomic (L190)).

The manuscript needs to be checked for consistency in how time is referenced (sometimes 6 ka BP, sometimes 6 ka). Also please explain what this means at the first reference.

Line-by-line specific comments:

L1: Please don't use LULC in the title, better to spell out what it is about.

L36: Unclear what is meant by "Current LULC scenarios". Is it current scenarios for the Holocene? Which part of the Holocene? Or, is it scenarios of LULC for the current climate (likely not, but it should be made more clear).

L42-45: From this it is unclear if the paper is just on evaluation of scenarios or if it is also about further refinement of the scenarios.

L44: What kind of "carbon-cycle simulations" are referred to here? Earth-system model simulations? Carbon cycle model simulations? Anything else?

L53-54: The new IPCC special report on land states that 70% of land is being influenced by anthropogenic activities. Is there a discrepancy here?

L56-57: Please define what is meant by "Mesolithic and Neolithic".

L79: "LULC change during the Holocene". It is unclear what is meant here. Is it over the full Holocene? Or, from any particular time in early or mid Holocene to any point during late Holocene (preindustrial?).

L189: "lack of uniform sampling through time" – does this include different national sampling strategies/resources for archeological excavations/sampling?

L190: What is taphonomic?

L331-343: Here, it is unclear whether the "already produced reconstructions" are products of REVEALS or if there are any other methods that have been involved.

L361: Suggest changing "observed climate" to "reconstructed climate".

L386-390: Here it is discussed changes in land use over time. The text gives the impression that there is always increasing land use with time "more conversion in earlier periods implies less conversion in later periods". Seems logical, but does this argument hold in a situation when land use is fluctuating with time (e.g. no land use – some land use – forest regrowth – no land use – again more land use …)?

L395: "to" missing after "due".

L440: How is land-use implemented in the models? Is it binary (i.e. 0 or 1) or fractional? In the latter case I guess that dynamical vegetation models could be used in combination with the land use information to derive vegetation type for the part of a gridbox not associated with land use.

L444-445: "free atmospheric CO2" needs a better explanation – for instance something like "…, allowing atmospheric CO2 concentrations to evolve in concert with fluxes to and from land and oceans".

L466: Please elaborate a bit on how good the assumption on "equilibrium" is for the Mid-Holocene? Was the carbon cycle (and climate) at equilibrium at that time?

L482-488: All references here are more than 10 years old. Are there no more recent studies of relevance?

Comments related to the Figures:

Figure 1: The color scale with the relatively dark green makes it difficult to see any of the rather small areas with land-use. It is difficult to understand why these two years have been chosen from the datasets (why not use the same reference year?). The font size at the color bar is too small.

Figure 2: The figure is difficult to read and it is not easy to see what is the final outcome of the scheme based on the figure. If it is something like "LULC scenario" I guess this should be something popping out on the right-hand side after going through the three steps in Phases 1-3. Also, it is not clear from the figure if there is any iterative part in the process where info is added to the scenarios based on constraints from phases 2-3? This could be better explained here and would also help to make the paper a bit more clear on a general level.

Figure 3. Here, font sizes are too small everywhere. What is SDPs? Please explain what the shading is for the maps (areas under human use?) and give a color bar. What are the circles in the lowermost panels?

Figure 4. Here is a box (Extensive/Minimal land use) that lacks some Level 2/3 information. Or it is redundant and can be removed? The labels on the land-use classes are quite specialized and several of the words are not everyday terms from my perspective (pastoralism, chinampas, taro pondfields, Peri-urban, Swidden). It would be good if these were a bit better explained, alternatively use different words). Also, why are there only Level 3 boxes for some of the Level 2 boxes?

Figure 5. This figure is not easily readable. The font size in the legends is way too small, the red dots in the upper panels are hardly distinguishable and the land-cover classes in the lowermost figure are not readable. Is the order left/right OK here? The figure indicates more people and land use at the earlier period (right panels) if I'm interpreting the figures correctly. In the figure caption "cal BC and BP" are used without definition anywhere. Also in the figure caption intervals defining the Middle and Early Neolothic time periods are given. Are these related to the more general statement on l56-57?

Figure 6. Realizing that these figures are conceptual they still need some better illustration. What are the different "squares" in the left panel second from the top? Gridsquares on a spatial map? Same question for the plots on the third row (and what is the bar with shading representing?)? Units lowermost left panel? Why is there a label "HYDE 3.x" on the top?

Figure 7. A suggestion here could be to remove the panel with the differences and make the other two a bit bigger and more easy to read (including larger font size on the color bar).

Figure 8. What are all the dots in the panels? Are the sites covering large areas? Biased to some regions? Evenly spread? Are all three panels for areas north of 30N? What are the associated uncertainty bars with the proxy-based data? With the models?

Comments on Table 1:

Why is "Modern" paleogeography and ice sheets used instead of "piControl"? And, how (if at all?) are these two differing? In the table "LC6k" is used supposedly for "LandCover6k", please spell out. What does it mean that pasture and crop distributions are "imposed"? I guess "imposed on top of the default vegetation in the 6ka experiment".

---

## Referee Comment (RC1) · Anonymous Referee #1 · 20 Nov 2019

This paper outlines planned work under the auspices of the LandCover6K working group to generate new estimates of Holocene land use/land cover (LULC) for use in climate model experiments. The proposed framework is novel in several regards, including explicit incorporation of archaeological information into the land cover estimates (e.g., 14C population estimates) and verification of the reconstruction using independent pollen networks and carbon cycle model simulations coupled with atmospheric CO2 estimates from ice cores.

I confess, this is a difficult manuscript to judge because it really is just a proposal. All these ideas outlined are great and, if executed, I believe will make significant contributions towards providing higher fidelity estimates of past LULC. The team assembled (assuming every on the author list is a full participant) is excellent, and certainly has the appropriate experience and set of skills. I guess the one thing that's lacking is really a "proof of concept", even for just a limited region. The authors mention some preliminary work that is being done, but this is not really discussed in any great detail. There's also a lack of any sort of timeline, research plan (who is funding this work? where will it be done? who will do what work?), or the current state of availability/processing of all the archaeological data that will be assembled to the new reconstruction.

At the end, i guess i'll recommend acceptance pending minor revisions. The proposed work sounds great, but it is completely unclear to me if this is the sort of paper GMD wants to publish.

---

## Referee Comment (RC2) · Anonymous Referee #2 · 25 Nov 2019

Review of Harrison et al., gmd-2019-125 Development and testing of scenarios for implementing Holocene LULC in Earth System Model Experiments

This paper describes a protocol for implementing LULC data in model simulations of Holocene climate as well as ways to create and test input to the LULC reconstructions and to test the output from the climate models. This is a welcome effort. LULC is a known climate driver that still is often not included in climate simulations. Proper data sets and protocol could hopefully make LULC a little bit easier to include in simulations in the future. Such a welcome effort will of course merit publication.

The paper describes a wide range of methods for a wide range of research areas. This

makes it a bit difficult for someone, like me, that is familiar with perhaps one or two of these research areas. I apologize already for the comments that I will raise that emanates from my ignorance in for example archaeology. But, this paper will be read by people that are not experts in everything that this paper covers; thus take my comments, although ignorant, as a motivation for rephrasing the text to be understandable for the non-expert.

The paper could use some rearrangements and clarifications before being published, otherwise the methods are sound and relevant. There are also some uncertainties about how all this will be done in practice. Comments follow below.

How are these LULC reconstructions better/different than HYDE and KK10? Are the methods different? Do we know that it is better? This may be obvious for everyone in the LULC business, but it is not explicitly explained in the text, at least not as far as I can see.

Is it possible to do uncertainty ranges? Some regions will inevitably be more uncertain than others. When you do a global map you tend to think that the uncertainties are the same everywhere. How do you deal with that? Also, the paper kind of assumes that data availability is as good as for the northern hemisphere in all of the world. I guess a lot of your methods won't work that well in parts of the world. How do you deal with that?

I think Section 2 is a bit confusing to follow. What is it that you want to show? Is it only to give a hint of the outline of the paper? That could be done much simpler. Section 1 introduces about the same concepts in a nice way, and the rest of the paper gives the details. It's hard to know if this is a description of the paper or something more general about the LandCover6k methodology (if these two are the same, please say so). I think that the rest of the paper will be easier to read if Section 2 clearly lists the three main points: 1) ways to improve data 2) ways to test data 3) the protocol. If this structure is kept and clear for the rest of the paper it will be easier to follow. Because it's mixture of

methods and results that is not always so easy to follow.

In Section 5 I don't get if REVEALS is used as an input to the LULC reconstructions or if it is used to evaluate the reconstruction. Is it only the fraction of open land that is evaluated? How is land cover reconstructed without REVEALS as the archaeological data (as I understand it) only give fraction of open land/land use.

For Section 6 I have a few concerns. First, should results be a part of a protocol paper? If it should, why are the results buried in the caption of Fig. 8? Are they old or new results? Make a proper paragraph explaining the results.

Second, the studies of LULC effects on simulated paleo climate that I'm familiar with tell clearly that despite radical changes in land cover the, although significant, differences in simulated climate are small compared to the uncertainty range in the proxies. It is not possible to assess which land-cover description is the most reasonable on the basis of a comparison of modelled climate with paleo climate reconstructions. (e.g. Strandberg et al., 2011; Strandberg et al., 2014). Your own results show this also. How do you plan to overcome this?

* Minor comments

L53: IPCC SRLUCC says 70% did you do a different kind of estimate? If you did, please explain why it's different.

L61: I don't think it's good to have the abbreviation LULC after the sentence "...as a result of land use". I guess LULC means land use and land cover. Spell out LULC before "affects the carbon cycle" on line 64 instead.

L95: "differences in the underlying assumptions" It would be interesting to know about what these assumptions are.

L175. "LULC scenarios" Is "scenarios" the right word here? I would go for "reconstruction" as "scenario" for me means an assumption about the future, with emphasis on the word assumption. These "LULC scenarios" are not based on assumptions but "a

number of products", i.e. they are in some way based on facts.

L229. "expert knowledge". How is "expert knowledge" done, is it even a method? Please explain and/or rephrase.

L281-295. Here, references to the different panels in Fig. 6 would be helpful.

L328-329. How is this done globally, is it possible to do on a global scale?

L332. "transient" and "500 years". Is it correct to call something with 500 year resolution transient? Or should it rather be time slices. Compare the use of "transient" in Section 8.

L405. "contributions to the land C inventory can be specified..." Is this possible to achieve? Your assumption builds on that.

L542-545. This is not possible without first improving proxy data.

Fig. 3 The text is far too small. No explanation for the grey shading or the white dots is given.

Fig. 4 Two boxes in Level 1 don't connect to Level 2. I can see that "No human land use" doesn't have to connect to Level 2, but is it then necessary to include it in the figure? I don't see how "Extensive/Minimal land use" fits in the picture.

Fig. 5 Too small legends.

Fig. 6 I don't understand the coupling between "LandCover 6k working group" and "HYDE 3.x". What does "→" mean? I don't understand many of the panels. What are the axes? What are the squares? What is the grey shading?

Fig. 7 Far too small legends.

Fig. 9 I don't understand this, but it seems to be more complicated than it sounds, but the surrounding text doesn't give much help.

Table 1 What does "Modern" mean here? If it is pre-industrial say so. If it is modern (=

20th century) explain why you don't use pre-industrial.

* References

Strandberg, G., Brandefelt, J., Kjellström, E. and Smith, B. 2011: High-resolution regional simulation of last glacial maximum climate over Europe. Tellus 63A, 107-125. DOI: 10.1111/j.1600-0870.2010.00485.x

Strandberg, G., Kjellström, E., Poska, A., Wagner, S., Gaillard, M.-J., Trondman, A.-K., Mauri, A., Davis, B. A. S., Kaplan, J. O., Birks, H. J. B., Bjune, A. E., Fyfe, R., Giesecke, T., Kalnina, L., Kangur, M., van der Knaap, W. O., Kokfelt, U., Kuneš, P., Latalowa, M., Marquer, L., Mazier, F., Nielsen, A. B., Smith, B., Seppä, H., and Sugita, S.: Regional climate model simulations for Europe at 6 and 0.2 k BP: sensitivity to changes in anthropogenic deforestation, Clim. Past, 10, 661-680, doi:10.5194/cp-10-661-2014, 2014.

---

## Author Comment (AC2) · 27 Nov 2019

The referee comments that it is difficult to judge the manuscript because it is a proposal for work to be done, although they recognise that the approach outlined is novel in several regards and would make a substantial contribution towards providing higher fidelity estimates of past LULC. We recognise that the paper is a somewhat unusual protocol in that it combines the development and testing of input data sets for model simulations as well as the description of the proposed simulations themselves. We have chosen to do this because we feel it is important that the palaeoclimate modellers who will be running these simulations understand the strengths and weaknesses of the input

data sets that are being developed. However, the ultimate goal here is to provide the protocol for simulations to be run by the PMIP group, building on the Holocene simulations that are already underway as part of CMIP6. The creation of the archaeological data sets and their use to improve LULC scenarios is currently being carried out by the PAGES LandCover6k working group, and it is anticipated that these data sets will be available for the PMIP community to use in 2020 – hence the need for a protocol to describe the planned experiments.

We can perhaps make the situation clearer by revising the introductory text to make it clear that work on the production of the input data sets is ongoing. (We will also be clarifying the status of individual components of the work in response to comments by Arneth and Kjellstrom). Specifically, we propose revising the final paragraph of the introduction to read:

The Past Global Changes (PAGES, http://www.pastglobalchanges.org/) LandCover6k Working Group (http://pastglobalchanges.org/landcover6k) is currently working to develop a rigorous and robust approach to provide data and data products that can be used to inform reconstructions of LULC (Gaillard et al., 2018). LULC changes are taken into account in simulations currently being made in the current phase of the Coupled Model Intercomparison Project (CMIP6) for the historic period and the future scenario runs (Eyring et al., 2016). They are also included in simulations of the past millennium (Jungclaus et al., 2017), in order to ensure that these runs mesh seamlessly with the historic simulations. However, the Land Use Harmonisation data set (LUH2: Hurtt et al., 2017) only extend back to 850 CE and thus LULC changes are currently not included in the CMIP6 palaeoclimate simulations, including mid-Holocene simulations, that are used as a test of how well state-of-the-art climate models reproduce large climate changes. In this paper, we discuss how archaeological data will be used to improve global LULC reconstructions for the Holocene. Given that there are large uncertainties associated with the primary data and further uncertainties may be introduced when this information is used to modify existing LULC scenarios, we
outline a series of tests that will be used to evaluate whether the revised scenarios are consistent with the changes implied by independent pollen-based reconstructions of land cover and whether they produce more realistic estimates of both carbon cycle and climate change. Finally, we present a protocol for implementing LULC in Earth System Model simulations to be carried out in the current phase of the Palaeoclimate Modelling Intercomparison Project (PMIP: Otto-Bleisner et al., 2017; Kageyama et al., 2018). However, the data sets and protocol will also be useful in later phases of other CMIP projects, including the Land Use Model Intercomparison Project (LUMIP) and the Land Surface, Snow and Soil Moisture Model Intercomparison Project (LS3MIP) (Lawrence et al., 2016; van den Hurk et al., 2016).

---

## Author Comment (AC3) · 28 Nov 2019

Response to comments by Almut Arneth

We thank Almut for her comments and suggestions. Comments in italics, response in normal script, suggested changes to text in bold.

...... one major aspect that seems missing from the approach. People need not only to eat, they also need to cook and heat, and to live. Has the group not discussed to -in addition to archaeological data- to also mine written historical records? This is probably most relevant for the last 1000+ years (rather than mid- Holocene), but surely

there can be assumptions about wood requirements for building materials (analogue to a per-capita area needed to be fed: how many people would live in an 'average' house/farm and how much would this would need), shipping fleets (records from ship-yards), charcoal making, furnaces for metal forging etc. I would imagine that at least in some regions this would have contributed perhaps already many centuries ago to deforestation. Could the authors comment on this aspect? We agree that wood harvesting is an important issue. Historical wood demand estimates have been made at a regional scale (e.g. McGrath et al., 2015) and indeed estimates of wood harvest are included in LUH2 (https://luh.umd.edu/data.shtml). However, there are very few direct estimates of wood consumption on the longer Holocene timescale that is the focus of the LandCover6k work. While it would be possible to implement approaches based on e.g. population estimates and assuming constant wood use per capita, this is unlikely to be more than a first approximation but a rigorous site-by-site evaluation of wood use through time across the globe though worthwhile would be very time-consuming. Thus, we envisage that the first round of Holocene LULC experiments would focus on the impacts of agricultural expansion and that gathering data to refine population-based estimates of wood harvest could be a future focus on the work of LandCover6k. However, since we agree that this is an important issue and we should make this clear, we propose to add a paragraph to the section describing the archae-ological data sources (line 255), as follows: The harvesting of wood for domestic fires, building, and for industrial activities such as transportation, pottery-making and metal-lurgy is an important aspect of human exploitation of the landscape in the pre-industrial period (McGrath et al., 2015). It has been argued that even Mesolithic hunter-gatherer communities shaped their environment through wood harvesting (Bishop et al., 2015). Approaches have been developed to quantifying the wood harvest associated with ar-chaeological settlements at specific times based on the evidence of types of wood use, household energy requirements, population size, and calorific value of the wood used (see e.g. Marston, 2009; Janssen et al., 2017). However, quantitative information on ancient technology and lifestyle is sparse and direct estimates of the amount of wood

harvest through time are likely to remain highly uncertain (Marston et al., 2017; Veal, 2017). Nevertheless, by combining modelling approaches with improved estimates of population size should allow changes in wood harvesting to be taken into account in LULC scenarios. Additional references Bishop, R.R., Church, M.J., Peter A. Rowley-Conwy, P.A., 2015. Firewood, food and human niche construction: the potential role of Mesolithic hunter–gatherers in actively structuring Scotland's woodlands. Quaternary Science Reviews, 108: 51-75. Janssen, E., Poblome, J., Claeys, J., Kint, V., Degryse, P., Marinova, E., Muys, B., 2017. Fuel for debating ancient economies. Calculating wood consumption at urban scale in Roman Imperial times. Journal of Archaeological Science: Reports 11: 592-599.

Marston, J.M., 2009. Modeling wood aquisition strategies from archaeological charcoal remains. Journal of Archaeological Science 36: 2192-2200. Marston, J.M., Holdaway, S.J., Wendrich, W., 2017. Early- and middle-Holocene wood exploitation in the Fayum basin, Egypt. The Holocene 27: 1812-1824. McGrath, M. J., Luyssaert, S., Meyfroidt, P., Kaplan, J. O., Burgi, M., Chen, Y., Erb, K., Gimmi, U., McInerney, D., Naudts, K., Otto, J., Pasztor, F., Ryder, J., Schelhaas, M. J., & Valade, A. (2015). Reconstructing European forest management from 1600 to 2010. Biogeosciences, 12(14), 4291-4316. doi:10.5194/bg-12-4291-2015 Veal, R., 2017. Wood and charcoal for Rome: towards an understanding of ancient regional fuel economics, In de Haas, T. & Gijs, T. (eds), Rural communities in a globalizing economy: new perspectives on the economic integration of Roman Italy, Brill, (New York and Leiden): pp.388-406.

Specific comments

Lines 63-65: For correctness, I would avoid using the term "feedback" here in the sense of change in process A affects process B, feeding back to A. LUC impacts on the carbon cycle are nothing more than an additional emission (or uptake), similar to other anthropogenic emissions, and the biophysical processes are related to albedo or ET change – but these are not feedbacks. We agree that this was not correctly phrased and will change this to: Direct climate impacts occur through changes in the surfaceenergy budget resulting from modifications of surface albedo, evapotranspiration, and canopy structure (biophysical impacts, e.g. Pongratz et al., 2010; Myhre et al., 2013; Perugini et al., 2017). LULC affects the carbon cycle through modifications in vegetation and soil carbon storage (biogeochemical impacts, e.g. Pongratz et al., 2010; Mahowald et al., 2017) and turnover times, which changes the C sink/source capacity of the terrestrial biosphere.

Lines 89-99: might be worth pointing out that the large discrepancies between Hyde and KK10 arise mostly from the assumptions about per-capita land requirements; to my knowledge their estimates of historical population changes through time (at least global totals) are more or less the same. We agree that we could be more explicit here and will change the sentence to read: However, differences in the underlying assumptions about land-use per capita, which are generalized from limited and often site-specific data, have resulted in large differences in the final reconstructions (Gaillard et al., 2010; Kaplan et al., 2017).

Lines 125-132: Given that these MIPs are already well under way, could you pls comment how realistic it is that the communities will be able to take up these protocols in time? Is it not more likely that the work will be most useful for many other studies that may not follow the tight schedule of the current AR6 MIP-frenzy, including work that would be useful also in context of the IPBES; and/or might feed into the next IPCC cycle ? Although the deadline for inclusion of material in the next IPCC Assessment report is looming, analyses of the CMIP6 simulations are not entirely tied to the current cycle and will continue after this year. Furthermore, the focus for most groups to date has been on Tier 1 type simulations and additional simulations will be made in the next years. This is certainly the case for the Palaeoclimate Modelling Intercomparison Project where, although the baseline mid-Holocene simulations are mostly completed, sensitivity simulations such as those we propose here will mostly not be started until 2020. However, we agree that we need to make it clear that the intention here is to provide a protocol for new model simulations. In order to do this,

and in response to comments by RC1, we have modified this paragraph as follows: The Past Global Change (PAGES, http://www.pastglobalchanges.org/) LandCover6k Working Group (http://pastglobalchanges.org/ini/wg/landcover6k) is currently working to develop a rigorous and robust approach to provide data and data products that can be used to inform reconstructions of LULC (Gaillard et al., 2018). LULC changes are taken into account in simulations currently being made in the current phase of the Coupled Model Intercomparison Project (CMIP6) for the historic period and the future scenario runs (Eyring et al., 2016). They are also included in simulations of the past millennium (Jungclaus et al., 2017), in order to ensure that these runs mesh seamlessly with the historic simulations. However, the Land Use Harmonisation data set (LUH2: Hurtt et al., 2017) only extend back to 850 CE and thus LULC changes are currently not included in the CMIP6 palaeoclimate simulations, including mid-Holocene simulations, that are used as a test of how well state-of-the-art climate models reproduce large climate changes. In this paper, we discuss how archaeological data will be used to improve global LULC reconstructions for the Holocene. Given that there are large uncertainties associated with the primary data and further uncertainties may be introduced when this information is used to modify existing LULC scenarios, we outline a series of tests that will be used to evaluate whether the revised scenarios are consistent with the changes implied by independent pollen-based reconstructions of land cover and whether they produce more realistic estimates of both carbon cycle and climate change. Finally, we present a protocol for implementing LULC in Earth System Model simulations to be carried out in the current phase of the Palaeoclimate Modelling Intercomparison Project (PMIP: Otto-Bleisner et al., 2017; Kageyama et al., 2018). However, the data sets and protocol will also be useful in later phases of other CMIP projects, including the Land Use Model Intercomparison Project (LUMIP) and the Land Surface, Snow and Soil Moisture Model Intercomparison Project (LS3MIP) (Lawrence et al., 2016; van den Hurk et al., 2016).

Lines 141/142: style; one 'required/requirements' might be sufficient. . .? We will change this to read: Generalising from site-specific data to landscape or regional

[Figure]

scales involves making assumptions about human behavior and cultural practices.

Figure 4: 'Wetland cultivation' in Level 3 – would that mean wetland drainage for agriculture? I assume it does, please clarify. The three categories under wet cultivations are: 1) creation of artificial wetlands for wetland crops, e.g., rice paddies, taro, 2) draining of wetlands in preparation for upland crops and pasture, e.g. polders or raised field systems, and 3) cultivation of existing wetlands (wetland cultivation). Thus, this last category does not mean drainage but rather preservation and use of existing wetlands. The terminology is explained in the cited Morrison et al. reference. However, we will modify the paragraph (lines 217-227) to make this clearer (and also to deal with comments made by Erik Kjellstrom), as follows: Maps of the distribution of archaeological sites or of areas linked to a given food production system have been produced for individual site catchments or small regions (e.g. Zimmermann et al., 2009; Barton et al., 2010; Kay et al., in press). LandCover6k is developing global land-use maps for specific time windows, based on a global hierarchical classification of land-use categories (Morrison et al., 2018) based on land-use types that are widely recognised from the archaeological record. At the highest level, the maps distinguish between areas where there is no (or only limited) evidence of land use, and areas characterized by hunting/foraging/fishing activities, pastoralism, agriculture, and urban/extractive land use (Fig. 4). Except in the cases where land use is minimal (no human land use, extensive/minimal land use), further distinctions are subsequently made to encompass the diversity of land-use activities in each land-use type (Fig. 4). A third level of distinction is made in the case of two categories (agroforestry, wet cultivation) where there are very different levels of intervention in different regions. Explanations of this terminology are given in Morrison et al. (2018). The LandCover6k land-use maps (see e.g. Fig. 5) will be based on different methods ranging from kernel-density estimates to expert knowledge depending on the quality and quantity of the archaeological information available from different regions.

Lines 146-162 – bit of an unspecific list, can be more precise, give more concrete examples? The aim of this text was to provide a general overview of the LandCover6k approach, to put subsequent sections into a broader context. each of the things listed are an explicit part of our strategy and thus further described. Since this obviously was not clear, we will revise the text and make explicit reference to the Figure describing the LandCover 6 scheme (Figure 2) and to the sections of the paper in which we develop each idea, as follows: Because of the inherent uncertainties, we advocate an iterative approach to incorporate archaeological data into LULC scenarios in LandCover6k (Fig. 2). We propose to revise the LULC scenario by incorporation of diverse archaeological inputs (Fig. 2, phase 1; see Sections 3 and 4) and to test the revised LULC scenarios for their plausibility and consistency with other lines of evidence (Fig. 2, phase 2 with iterative testing; see Sections 5-7). As a first test, the revised LULC scenarios of the extent of cropland and grazing land through time will be compared with independent data on land-cover changes, specifically pollen-based reconstructions of the extent of open land (see e.g. Trondman et al., 2015; Kaplan et al., 2017) (Section 5). Further testing the LULC scenarios involve sensitivity tests using global climate models (Section 6) and global vegetation-carbon cycle models (Section 7). While the computational cost of the climate simulations can be minimized using equilibrium time-slice simulations, the carbon cycle constraint relies on transient simulations, but may be derived from uncoupled, land-only simulations. Simulated climates at key times can be evaluated against reconstructions of climate variables (e.g. Bartlein et al., 2011) (Section 6). The parallel evolution of CO2 and its isotopic composition ($\delta$13C) can be used to derive the carbon balance of the terrestrial biosphere and the ocean separately (Elsig et al., 2009) and, in combination with estimates for other contributors to land carbon changes such as C sequestration by peat buildup, provides a strong constraint on the evolution of LULC through time. An under- or over-prediction of anthropogenic LULC-related CO2 emissions during a specific interval results in consequences for the dynamics of the atmospheric greenhouse gas burden in subsequent times (Stocker et al., 2017) (Section 7). Thus, these tests can be used to identify issues in the original archaeological datasets and/or the way these data were incorporated into the LULC scenarios

that require further refinement. Phase 3 of the protocol (Fig. 2) proposes specific implementation of the revised LULC in Earth System Model simulations (Section 8).

We will also modify the caption to Figure 2 as follows: Figure 2: Proposed scheme for developing robust LULC scenarios through iterative testing and refinement, as input to Earth System Model (ESM) simulations. The archaeological inputs developed in Phase 1 can be used independently or together to improve the LULC reconstructions (Phase 2); iterative testing of the LULC scenario reconstruction (phase 2) will ensure that these inputs are reliable before they are used of ESM simulations (phase 3). The uppermost three LULC simulations capitalize on already planned baseline simulations without LULC; the lowermost two simulations are envisaged as new sensitivity experiments.

Section 3.1 – this section wasn't entirely clear to me. What samples are we talking about exactly, what is being dated, where do the samples come from? Could you provide an illustrative example? We will modify this section to make it clearer that we are referring to dated archaeological material, as follows: Radiocarbon is the most routinely used absolute dating technique in archaeology, especially for the Holocene. Many thousands of radiocarbon dates on archaeological material are available from the literature. A number of regional and pan-regional initiatives are compiling these records through exhaustive survey of the archaeological literature (e.g. the Canadian Archaeological Radiocarbon Database: https://www.canadianarchaeology.ca/).

We will also modify the text describing the sources of bias: There are biases that could affect the expected one-to-one relationship between number of people and number of radiocarbon dates on archaeological material, including lack of uniform sampling through time and space caused by different archaeological research interests and traditions in different regions) and increased preservation issues with increasing age.

Since there are several different ways this approach is being applied, we do not feel a single illustrative example would be adequate. We will therefore modify the final sentence of this paragraph to indicate that the references given refer to specific regional examples, as follows: Radiocarbon dates have been successfully used in several regions to identify population fluctuations associated with the introduction of farming and subsequent changes in farming regimes (western Europe: Shennan et al., 2013; Wyoming: Zahid et al., 2016; South Korea: Oh et al., 2017; see also Freeman et al., 2018) as well as climatic oscillations (Ireland: Whitehouse et al., 2014; Japan: Crema et al., 2016).

Figure 5: I liked the Figure, is nice to see a concrete, illustrative example of the planned approach. However, it was not entirely obvious to me what the top and bottom panels in Fig. 5 are meant to convey: is it to show the improvements that can be made by adding the new information to the existing LandCover 6a? Or what is exactly the added value of the two combined? And what's the reasoning behind the 10-15% and the 5% mentioned in lines 269/270? This figure illustrates alternative approaches to mapping land use, with the upper panels showing the distribution of archaeological sites and how these data are generalised to an provide an estimate of the extent of land use. The lower panels show the same data but superimposed on the land use classification scheme used by LandCover6k. It is unrealistic for these periods - or even today - to consider that the entire 64km2 is continuously covered with fields, and the percentages given are estimates of how much of each grid cell was being used in cells assigned to low-level agriculture in different parts of Ireland. We will modify the caption to make this clearer, as follows: An example of regional land-use mapping. The upper panels show the distribution of known archaeological sites superimposed on kernel density estimates of the extent of land-use based on the density of sites, and the lower panels show these data superimposed on the LandCover6k land-use classes for the Middle Neolithic (3600-3400 cal BC, 5600-5400 BP) (left panels) and the Early Neolithic (3750-3600 cal BC, 5750-5600 BP) (right panels) of Ireland. Data points derive from 14C dated archaeological sites and distributions of settlements and monuments that have been assigned to each archaeological period following the dataset published in McLaughlin et al. (2016). The assigned land-use classes are inferred from archaeological material from one (or more) sites within the grid box. It should not be assumed that

the whole gridcell was being used for agriculture during the Middle and Early Neolithic. Informed assessment suggests that agricultural land (crop growing and grazing, combined) probably occupied between 10-15% of the total grid area in the low-level food production regions of the eastern and western coastal areas, whilst agricultural land likely represents 5% or less of the total grid cell area in inland areas.

Lines 288/289: how do you obtain information about past irrigation? From archaeological data (irrigation structures?) I assume? Likewise, per-capita land needs surely change over time, agreed. But how can these estimates be obtained, could you provide more explanation and/or references to methods as to how to do this? We will expand the text to clarify these points, as follows: Information on the extent of rain-fed versus irrigated agriculture, as indicated by the presence of irrigation structures associated with archaeological sites, can also be used to refine the distribution of these classes in the LULC scenarios. Per-capita land-use estimates and their changes through time (see e.g. Hughes et al., 2018; Weiberg et al., 2019) provide a further refinement of the LULC scenarios, allowing a better characterization of the distinction between e.g. areas given over to extensive versus intensive animal production (rangeland versus pasture in the HYDE 3.2 terminology).

Additional references Weiberg, E., Hughes, R. E., Finné, M., Bonnier, A., & Kaplan, J. O. (2019). Mediterranean land use systems from prehistory to antiquity: a case study from Peloponnese (Greece). Journal of Land Use Science, 1-20. doi:10.1080/1747423x.2019.1639836 Hughes, R., Weiberg, E., Bonnier, A., Finné, M., & Kaplan, J. (2018). Quantifying land use in past societies from cultural practice and archaeological data. Land, 7(1), 9. doi:10.3390/land7010009

Figure 6, just for illustrative purpose only: the panels 'land use classification input' and 'revised land use allocation' look identical, might be illustrative to not only change the legend but also the drawing. These two panels necessarily look identical because the archaeological data shown in the lefthand panel are explicitly incorporated into the scenario. Unlike in the other examples, it is difficult to show the before/after situation

here. However, we can expand the caption to make this clearer, as follows: Schematic illustration of the proposed implementation of 14C-based population estimates, date of first agriculture, land-use maps, and land-use per capita information in the HYDE model (here indicated as HYDE3.x). The archaeological data are represented as values for a grid cell in geographic space at a given time for date of first agriculture and land use, but as a time series for a specific grid cell for population and land-use per capita. In the case of population estimates, date of first agriculture and land-use per capita data, we show the initial estimate and the revised estimate after taking the archaeological information into account in the HYDE3.x plot. It should be assumed in the case of the land-use mapping that the original estimate was that there was no land use in this region.

Line 327-329: what's the basis for the optimism that 'eventually' these pollen-based re-constructions will also be available elsewhere (presumably: the tropics), is there initial work that points in that direction? And what's the pros/cons of the "other" pollen-based reconstructions that are mentioned? There is indeed work going on the collect RPP data in other parts of the world, and we will expand the text to explain this and to ex-plain the pros/cons of the other techniques, as follows: The REVEALS approach has been used to reconstruct changes in the amount of open land through time across the northern extratropics (Figure 7; Dawson et al., 2018) through the Holocene with a time resolution of 500 years from 11.5ka to 0.7ka BP, and three historical time windows (modern–0.1ka BP, 0.1–0.35ka BP, and 0.35–0.7ka BP). A major limitation in applying REVEALS globally is requirement for information about the relative pollen productivity (RPP) of individual pollen taxa, which is currently largely lacking for the tropics. How-ever, LandCover6k has been collecting RPPs for China, South-East India, Cameroon, Brazil and Argentina and pollen-based land-cover reconstructions will be available for at sufficient parts of the tropics to allow testing of the scenarios. Another limitation of REVEALS estimates is that RPP estimates are available for cultivated cereals but not for other cultivars or cropland weeds, so the LandCover6k reconstructions will generally underestimate cropland cover (Trondman et al., 2015). It may also be possible to use

alternative pollen-based reconstructions of land cover changes, such as the Modern Analogue Approach (MAT: e.g. Tarasov et al., 2007; Zanon et al. 2018); pseudo-biomization (e.g. Fyfe et al., 2014) or STEPPS (Dawson et al., 2016). While none of these methods require RPPs, MAT and STEPPS can only be applied in regions where the pollen datasets have dense coverage (such as Europe and North America) and pseudo-biomization is affected by the non-linearity of the pollen-vegetation relationship that the REVEALS approach is designed to remove.

Lines 385/386: "known" today is not quite true unfortunately. There are still sizeable discrepancies in today's land cover estimates in terms of major classes such as crop-land, pasture, forest, 'other' (let alone in the degree to which these are being used). Partially this arises from disagreements in terms of how a pasture or forest is defined. There is no need to add a long discussion but pls. revise the sentence slightly to express that there is also uncertainty for today. We agree that this statement was a little too optimistic and will change the text to read: First, reconstructions of the total land under agricultural use must converge on the present-day state, which is relatively well constrained by satellite land-cover observations and national statistics on the amount of land under use.

Lines 383-399: The scaling aspect is important. However, cumulative LUC C emissions differ substantially depending on whether "net" or "gross" area changes are being cal-culated. The total agricultural area might be the same in both approaches, but the 'gross' approach considers expansion and reduction that might occur within a gridcell. The most prominent example is shifting cultivation, and today is mostly restricted to tropical regions. However, others have pointed out that such gross transition of course also are relevant on other parts of the world (see e.g., Fuchs et al., GCB, 2015), and were possibly even more so further back in time. The challenges that arise from this aspect are mentioned later in the Outcomes section but I wonder if it's not better to introduce these already here. We agree that it would be important to account for the difference, and this is one reason that we discuss this issue in the Outcomes section

(lines 532 et seq.). Unfortunately, the only way to do this globally at the present time is by making assumptions about farming practices (e.g. how much land is abandoned or fallowed in a given year). The archaeological record does not provide a very strong basis for quantifying this. We will modify the text describe the carbon-cycle simulations to clarify that these simulations will necessitate making assumptions about the nature of land-use turnover, as follows: Transient simulations with a model that simulates $CO_2$ emissions in response to anthropogenic LULC can be used to test the reliability of the LULC changes through time, by comparing results obtained with prescribed LULC changes through time against a baseline simulation without imposed LULC. This will necessitate making informed decisions about the fraction of land under cultivation that is abandoned or left fallow each year, and the maximum extent of land affected by such episodic cultivation. The simulations will be driven by climate outputs (temperature, precipitation and cloud cover) from an existing existing transient climate simulation made with the ECHAM model (Fischer and Jungclaus, 2011) and $CO_2$ prescribed from ice-core records. The $CO_2$ emission estimates from these two simulations will then be evaluated using C budget constraints. This evaluation will allow us to pinpoint potential discrepancies between known terrestrial C balance changes and estimated LULC $CO_2$ emission in given periods over the Holocene.

---

## Author Comment (AC5) · 28 Nov 2019

Response to reviewer RC2 Comments in italics, response in normal script, suggested changes to text in bold. We note that several of these comments are similar to those posted by Erik KjellstroÌLm, and in these cases we have already responded and note this here.

How are these LULC reconstructions better/different than HYDE and KK10? Are the methods different? Do we know that it is better? This may be obvious for everyone in the LULC business, but it is not explicitly explained in the text, at least not as far as I can see. The LULC reconstructions we are proposing will be refinements of HYDE and

KK10 that take account of a wider range of archaeological data. We describe these data in Section 3 and how they will improve the current HYDE and KK10 scenarios in section 4. In response to comments by the other reviewers, we have expanded the text in both of these sections to be more explicit about the data and how these data will be incorporated into the existing scenarios. The main improvements hinge on having better estimates of population changes based on the density of archaeological settlement evidence, better information for the initiation of agriculture in a region, more regionally specific information about land use, and more nuanced information about land-use per capita than the somewhat generic estimates used in KK10 or the single global assumption about land-use per capita that underpins HYDE. Until these data are used to revise the scenarios, and tested against independent data (as described in Sections 5, 6 and 7), we cannot be sure what impact they will have. Our contention is that it is surely better to incorporate information about human exploitation of the landscape than to rely on estimates that we know are based on relatively simple assumptions and which, in any case, differ markedly from one another as a consequence of these assumptions. We will take the opportunity to make a clearer statement about this in our final outcomes and perspective section, as follows: LandCover6k has developed a protocol for using archaeological information to improve existing scenarios of LULC changes during the Holocene, specifically by using archaeological data to provide better estimates of regional population changes through time, better information on the date of initiation of agriculture in a region, more regionally specific information about the type of land use, and more nuanced information about land-use per capita than currently implemented in the LULC scenarios generated by HYDE and KK10. While the final global archaeological data sets are still in production, fast-track priority products have been created and their impact on current LULC scenarios is being tested.

Is it possible to do uncertainty ranges? Some regions will inevitably be more uncertain than others. When you do a global map you tend to think that the uncertainties are the same everywhere. How do you deal with that? Also, the paper kind of assumes that data availability is as good as for the northern hemisphere in all of the world. I guess

a lot of your methods won't work that well in parts of the world. How do you deal with that? We are fully aware that the amount and quality of the archaeological data inputs is not the same everywhere, and indeed we state this in our outcomes and perspective section (line 512 et seq.). Nevertheless, incorporating information from regions where the data is good and identifying regiona where there is less certainty will certainly go some way to improving the scenarios. It should be remembered that the archaeological itself is only input to the scenarios and that both HYDE and KK10 interpolate these data to generate global scenarios of land use. It is certainly possible an our intention to provide uncertainty ranges on the estimates (see for e.g. the caption to Figure 5). These can be used to generate for example high-end and low-end scenarios of LULC change, a practice that parallels the implementation of LULC changes in future simulations. We did not spell this out clearly in the paper, and so we will take the opportunity to do so, as follows: Although the work of LandCover6k will provide more solid knowledge about anthropogenic modification of the landscape, some information will inevitably be missing and some key regions will be poorly covered. There will still be large uncertainties associated with LULC scenarios. Documenting these uncertainties is an important goal of the LandCover6k project, and will allow the generation of multiple scenarios comparable to the "low-end", "high-end" scenarios used for e.g. in future projections. Furthermore, we have proposed a series of tests that will help to evaluate the realism of the final scenarios, based on independent evidence from pollen-based reconstructions of land cover, reconstructions of climate, and carbon-cycle constraints. These tests should help in identifying which of the potential LULC reconstructions are most realistic and constraining the sources of uncertainty.

I think Section 2 is a bit confusing to follow. What is it that you want to show? Is it only to give a hint of the outline of the paper? That could be done much simpler. Section 1 introduces about the same concepts in a nice way, and the rest of the paper gives the details. It's hard to know if this is a description of the paper or something more general about the LandCover6k methodology (if these two are the same, please say so). I think that the rest of the paper will be easier to read if Section 2 clearly lists the

three main points: 1) ways to improve data 2) ways to test data 3) the protocol. If this structure is kept and clear for the rest of the paper it will be easier to follow. Because it's mixture of methods and results that is not always so easy to follow. This Section was designed to explain the methodology we are using and in particular the different phases of work. within the protocol. In response to comments by Almut Arneth we propose to revise this section to make it clearer about the three different phases of work outlined in this protocol, i.e. (a). using archaeological data to refine LULC scenarios, (b) testing the revised scenarios and (c) running climate model simulations to examine the impact of LULC changes on climate, as follows: Because of the inherent uncertainties, we advocate an iterative approach to incorporate archaeological data into LULC scenarios in LandCover6k (Fig. 2). We propose to revise the LULC scenario by incorporation of diverse archaeological inputs (Fig. 2, phase 1; see Sections 3 and 4) and to test the revised LULC scenarios for their plausibility and consistency with other lines of evidence (Fig. 2, phase 2 with iterative testing; see Sections 5-7). As a first test, the revised LULC scenarios of the extent of cropland and grazing land through time will be compared with independent data on land-cover changes, specifically pollen-based reconstructions of the extent of open land (see e.g. Trondman et al., 2015; Kaplan et al., 2017) (Section 5). Further testing the LULC scenarios involve sensitivity tests using global climate models (Section 6) and global vegetation-carbon cycle models (Section 7). While the computational cost of the climate simulations can be minimized using equilibrium time-slice simulations, the carbon cycle constraint relies on transient simulations, but may be derived from uncoupled, land-only simulations. Simulated climates at key times can be evaluated against reconstructions of climate variables (e.g. Bartlein et al., 2011) (Section 6). The parallel evolution of $CO_2$ and its isotopic composition ($\delta13C$) can be used to derive the carbon balance of the terrestrial biosphere and the ocean separately (Elsig et al., 2009) and, in combination with estimates for other contributors to land carbon changes such as C sequestration by peat buildup, provides a strong constraint on the evolution of LULC through time. An under- or over-prediction of anthropogenic LULC-related $CO_2$ emissions during a

specific interval results in consequences for the dynamics of the atmospheric greenhouse gas burden in subsequent times (Stocker et al., 2017) (Section 7). Thus, these tests can be used to identify issues in the original archaeological datasets and/or the way these data were incorporated into the LULC scenarios that require further refinement. Phase 3 of the protocol (Fig. 2) proposes specific implementation of the revised LULC in Earth System Model simulations (Section 8). In Section 5 I don't get if REVEALS is used as an input to the LULC reconstructions or if it is used to evaluate the reconstruction. Is it only the fraction of open land that is evaluated? How is land cover reconstructed without REVEALS as the archaeological data (as I understand it) only give fraction of open land/land use. The REVEALS reconstructions are being used here as a way of evaluating the LULC reconstructions derived from archaeological information. REVEALS reconstructions could be used as input to the LULC scenarios, especially in regions where the archaeological information is sparse, but as we explain in the text (lines 333-339) there are problems in doing this because (a) pollen-based reconstructions cannot distinguish between anthropogenic and climatically determined natural open land (e.g. natural grasslands, steppes, wetlands) and (b) REVEALS underestimates cropland cover because there are no RPP estimates for cultivars other than cereals. In contrast, the archaeological data provides information on different types of agriculture (crops versus grazing versus mixed) and the types of crops being grown, direct information on the area affected and indirect estimates of the land-use per capita associated with different types of agriculture at different times that can be used to infer the area being used. However, since there is some confusion about the different information obtained from the two different sources and how we will use the REVEALS data for evaluation we will expand the text to explain this procedure more explicitly, as follows: Pollen-based vegetation reconstructions can be used to corroborate archaeological information on the date of first agriculture from the appearance of cereals and agricultural weeds. These reconstructions can also be used to test the LULC reconstructions, either using relative changes in forest cover or reconstructions of the area occupied by different land cover types. LandCover6k uses the REVEALS

model (Sugita, 2007) to estimate vegetation cover from fossil pollen assemblages. The REVEALS model predicts the relationship between pollen deposition in large lakes and the abundance of individual plant taxa in the surrounding vegetation at a large spatial scale (ca. 100 km x 100 km; Hellman et al., 2008a, b) using models of pollen dispersal and deposition. REVEALS can also be used with pollen records from multiple small lakes or peat bogs (Trondman et al., 2016) although this results in larger uncertainties in the estimated area occupied by individual taxa. The estimates obtained for individual taxa are summed to produce estimates of the area occupied by either plant functional (e.g. summer-green trees, evergreen trees) or land cover (e.g. open land, grazing land, cropland) types. We will also add a final sentence to this section as follows: However, overestimation of the area of open land in the LULC scenarios might suggest problems either in the archaeological inputs or their implementation, especially for times or regions when other evidence indicates cereals were the major crop. In this sense, despite potential problems, the LandCover6k pollen-based reconstructions of land cover will provide an important independent test of the revised LULC scenarios.

For Section 6 I have a few concerns. First, should results be a part of a protocol paper? If it should, why are the results buried in the caption of Fig. 8? Are they old or new results? Make a proper paragraph explaining the results. Section 6 is describing our approach for evaluating the new LULC scenarios by seeing whether they have an impact on simulated climate, and whether this impact is to produce a better a better simulation of climate or not. We illustrate this approach by showing two existing simulations, one with and one without LULC changes. The simulations are published and we cite this publication (Smith et al., 2016). It is not our intention here to comment on the simulations themselves, simply to illustrate how we would evaluate new simulations. We can clarify this by modifying the caption, as follows: Quantitative comparison of the change in climate between the mid-Holocene (6ka) and the pre-industrial period as shown by pollen-based reconstructions (from Bartlein et al., 2011) and in simulations with and without the incorporation of land-use change (from Smith et al., 2016). This figure illustrates the approach that will be taken to evaluate the impact of new LULC scenarios

Interactive
comment

on climate. The imposed land-use changes at 6ka were derived from the KK10 scenario (Kaplan et al., 2011). The plots show comparisons of mean annual temperature (MAT), mean temperature of the coldest month (MTCO) and mean annual precipitation (MAP) for the northern extratropics (north of 30° N). Although the incorporation of land use produces somewhat warmer and wetter climates in these simulations, overall the incorporation of land-use produces no improvement of the simulated climates at sites with pollen-based reconstructions. Second, the studies of LULC effects on simulated paleo climate that I'm familiar with tell clearly that despite radical changes in land cover the, although significant, differences in simulated climate are small compared to the uncertainty range in the proxies. It is not possible to assess which land-cover description is the most reasonable on the basis of a comparison of modelled climate with paleo climate reconstructions. (e.g. Strandberg et al., 2011; Strandberg et al., 2014). Your own results show this also. How do you plan to overcome this? The Smith et al. simulations show regional changes in summer temperature (JJA) due to LULC of 2-3 degrees C in e.g. North America, Europe and China in the late Holocene, and changes of the same magnitude for more limited regions in the early Holocene. This is certainly within the detection range of the pollen-based reconstructions of summer temperature for these regions. Thus, we are sure that such comparisons will be a useful additional assessment of the new LULC simulations. In fact, in the Smith et al. simulations shown in Figure 8 to illustrate our approach, show an improvement in simulated climate in the high latitudes (increased warming) that is offset in this comparison by a degradation in simulated climate elsewhere. Thus, in our evaluations of the impact LULC on simulated climate we will necessarily have to make more detailed regional comparisons – and this will be useful information for the diagnosis of the improved LULC simulations because it might pinpoint regions where the imposed LULC is wrong. We have already modified this paragraph in response to comments by KjellstroÌLm to clarify this point, as follows: A second test of the realism of the improved LULC scenarios is to examine whether incorporating LULC changes improves the realism of the simulated climate when compared to palaeoclimate reconstructions

(Figure 8). The mid-Holocene (6000 years ago, 6ka BP) is an ideal candidate for such a test because benchmark data sets of quantitative climate reconstructions are available (e.g. Bartlein et al., 2011), the interval has been a focus through multiple phases of PMIP and control simulations with no LULC have already been run, and evaluation of these simulations has identified regions where there are major discrepancies between simulated and observed climates e.g. the observed expansion of northern hemisphere monsoons, climate changes over Europe, the magnitude of high-latitude warming, and wetter conditions in central Eurasia (Mauri et al., 2014; Harrison et al., 2015; Bartlein et al., 2017). There are discernible anthropogenic impacts on the landscape in many of these regions by 6 ka, although they are not as strong as during the later Holocene and they are not present everywhere. Nevertheless, the 6ka BP interval provides a good focus for testing improvements to the LULC scenarios. Such an evaluation would need to go beyond the global comparison made here (Figure 8) to regional comparisons to identify whether improvements in regions where there is a large anthropogenic impact on land cover do not result in a degradation in the simulated climate elsewhere.

Minor comments L53: IPCC SRLUCC says 70% did you do a different kind of estimate? If you did, please explain why it's different. To clarify, the estimate we provide is taken from the cited references. It is obviously difficult to provide an overall estimate of how much of the land surface is affected by human activities because it depends on whether the focus is on direct appropriation for agriculture resulting in a fundamental change in land cover or whether any anthropogenic influence is being taken into account. In fact, the Land Report states (section 1.1.2.2) that between 60–85% of the total forested area and between 72-89% of non-forested land is used, but it also makes it clear that the level of usage is variable with only 10% being intensively managed, two-thirds being moderately managed and the remainder at low intensities. Only about one third of the used land is associated with changed land cover. The Report states that differences in definitions and lack of information about management practice means that the estimates of human usage are uncertain. So, in this sense our statement is compatible with the Land Report, in that the estimated 40% refers to the

area being used for agriculture and we go on to say that large parts of the rest of the land area are being influenced in some way by human activities. However, our point here is not to quantify the extent of use but simply to point out that there is considerable anthropogenic modification on the landscape globally. We will acknowledge the work of the Land Report – which came out after we submitted this paper – and modify this sentence as follows: Today, ca 10% the ice-free land surface is estimated to be intensively managed and much of the reminder is under less intense anthropogenic use or influenced by human activities (Arneth et al., 2019).

L61: I don't think it's good to have the abbreviation LULC after the sentence "...as a result of land use". I guess LULC means land use and land cover. Spell out LULC before "affects the carbon cycle" on line 64 instead. The sentence currently reads "changes in land cover as a result of land use (LULC)". We can expand this as follows: .... changes in land cover as a result of land use (land use land cover: LULC)

L95: "differences in the underlying assumptions" It would be interesting to know about what these assumptions are. We agree that we could be more explicit here and will change the sentence to read: However, differences in the underlying assumptions about land-use per capita, which are generalized from limited and often site-specific data, have resulted in large differences in the final reconstructions (Gaillard et al., 2010; Kaplan et al., 2017).

L175. "LULC scenarios" Is "scenarios" the right word here? I would go for "reconstruction" as "scenario" for me means an assumption about the future, with emphasis on the word assumption. These "LULC scenarios" are not based on assumptions but "a number of products", i.e. they are in some way based on facts. The term scenario is indeed used to describes a trajectory of change in the future based on making assumptions about e.g. behaviour patterns. It can equally well be used to apply to the past LULC changes which may be informed to some extent by data but are also underpinned by assumptions. Indeed, as we point out (see response above) it is these assumptions that give rise to the very large differences between the different "products" currently

available. We do not claim that incorporating archaeological information will change the basis for scenario-creation; merely that incorporating more data that will help refine these assumptions, the resulting scenarios will become more realistic.

L229. "expert knowledge". How is "expert knowledge" done, is it even a method? Please explain and/or rephrase. There are some regions where there are very few archaeological sites and where statistical methods are therefore difficult to apply. In such regions, we will be forced to use the insights of the archaeologists who worked on the sites about what kind of land use the archaeological records imply. We feel that this is more informative than leaving grid cells blank. We will change the sentence to read: The LandCover6k land-use maps (see e.g. Fig. 5) will be based on different methods ranging from kernel-density estimates to expert assessments depending on the quality and quantity of the archaeological information available from different regions.

L281-295. Here, references to the different panels in Fig. 6 would be helpful. We will modify the figure to add labels so that we refer to the separate panels in the text.

L328-329. How is this done globally, is it possible to do on a global scale? It is not necessary to have global reconstructions to evaluate LULC scenarios, although this is of course desirable. The ultimate goal of PAGES LandCover6k is to provide such reconstructions globally, and we explain that lack of tropical RPPs is the current limitation on providing a global reconstruction using REVEALS. As we point out in our response to a comment by Almut Arneth about the likelihood of having global reconstructions, LandCover6k has been collecting tropical RPPs which will thus facilitate global reconstructions. Furthermore, as we point out in the paper, there are alternative methods that have been used in regions where there are no RPPs and these reconstructions can also be used to evaluate the LULC scenarios. We have expanded the text describing the pollen-based reconstructions (in response to Almut's comments), as follows: The REVEALS approach has been used to reconstruct changes in the amount of open land through time across the northern extratropics (Figure 7; Dawson et al., 2018) through the Holocene with a time resolution of 500 years from 11.5ka to 0.7ka BP,

and three historical time windows (modern–0.1ka BP, 0.1–0.35ka BP, and 0.35–0.7ka BP). A major limitation in applying REVEALS globally is requirement for information about the relative pollen productivity (RPP) of individual pollen taxa, which is currently largely lacking for the tropics. However, LandCover6k has been collecting RPPs for China, South-East India, Cameroon, Brazil and Argentina and pollen-based land-cover reconstructions will be available for at sufficient parts of the tropics to allow testing of the scenarios. Another limitation of REVEALS estimates is that RPP estimates are available for cultivated cereals but not for other cultivars or cropland weeds, so the LandCover6k reconstructions will generally underestimate cropland cover (Trondman et al., 2015). It may also be possible to use alternative pollen-based reconstructions of land cover changes, such as the Modern Analogue Approach (MAT: e.g. Tarasov et al., 2007; Zanon et al. 2018); pseudo-biomization (e.g. Fyfe et al., 2014) or STEPPS (Dawson et al., 2016). While none of these methods require RPPs, MAT and STEPPS can only be applied in regions where the pollen datasets have dense coverage (such as Europe and North America) and pseudo-biomization is affected by the non-linearity of the pollen-vegetation relationship that the REVEALS approach is designed to re-move. L332. "transient" and "500 years". Is it correct to call something with 500 year resolution transient? Or should it rather be time slices. Compare the use of "transient" in Section 8. It is true that in a modelling context we use the term transient to mean "every year" whereas the pollen-based reconstructions are currently snapshots at 500 year intervals, except in the last millennium. It would be possible to provide recon-structions at finer intervals, for example at 50 year intervals subject to the sampling resolution and the uncertainty of the age model of the individual pollen cores. We will modify the wording here to differentiate between the model simulations and the pollen-based reconstructions, as follows: LandCover6k has already produced reconstructions for the northern extratropics. These reconstructions provide snapshots through the Holocene with a time resolution of 500 years until 0.7ka BP, and three historical time windows (modern–0.1ka BP, 0.1–0.35ka BP, and 0.35–0.7ka BP). L405. "contributions to the land C inventory can be specified..." Is this possible to achieve? Your assumption
builds on that. The main independent contribution to the land C inventory is the build up of peat through the Holocene and this is, at least to first order, known from syntheses of peat records. We can expand this text to be more specific, as follows: Providing that all of the natural contributions to the land C inventory (e.g. the build up of natural peatlands: Loisel et al., 2014) can be specified from independent evidence, the anthropogenic sources can be estimated as the difference between the total terrestrial C budget and natural contributions (Figure 9) at any specific time.

Additional reference Loisel J, Yu Z, Beilman DW, Camill P, Alm J, Amesbury MJ, Anderson D, Andersson S, Bochicchio C, Barber K, Belyea LR, Bunbury J, Chambers FM, Charman DJ, Vleeschouwer FD, Fiałkiewicz-Kozieł B, Finkelstein SA, Gałka M, Garneau M, Hammarlund D, Hinchcliffe W, Holmquist J, Hughes P, Jones MC, Klein ES, Kokfelt U, Korhola A, Kuhry P, Lamarre A, Lamentowicz M, Large D, Lavoie M, MacDonald G, Magnan G, Mäkilä M, Mallon G, Mathijssen P, Mauquoy D, McCarroll J, Moore TR, Nichols J, O'Reilly B, Oksanen P, Packalen M, Peteet D, Richard PJ, Robinson S, Ronkainen T, Rundgren M, Sannel ABK, Tarnocai C, Thom T, Tuittila E-S, Turetsky M, Väliranta M, Linden Mvd, Geel Bv, Bellen Sv, Vitt D, Zhao Y & Zhou W, 2014. A database and synthesis of northern peatland soil properties and Holocene carbon and nitrogen accumulation. The Holocene 24: 1028-1042

L542-545. This is not possible without first improving proxy data. We do not understand this comment. The point of this protocol paper is to explain how we will improve the land use scenarios so that they can be used to drive model simulations. The point here is that these experiments could be used to explore whether the land-use changes are implicated in e.g. abrupt events or whether specific land-use changes associated with population changes used in the scenarios produce significant effects on climate. Fig. 3 The text is far too small. No explanation for the grey shading or the white dots is given. A similar point was raised by KjellstroÌ Lm and we have expanded the text and modified the caption to explain this figure better

Fig. 4 Two boxes in Level 1 don't connect to Level 2. I can see that "No human land use"

doesn't have to connect to Level 2, but is it then necessary to include it in the figure? I don't see how "Extensive/Minimal land use" fits in the picture. As we have said in our response to KjellstroÌLm, the Figure is included for illustrative purposes and shows the scheme of land-use categories developed by LandCover6k to be used by the archaeological community to map land-use in different regions of the world. The terminology is that used to describe different kinds of agriculture by archaeologists, and there is a handbook (which we can refer to) that defines these terms. As we explain in the text, these land-use types will have to be translated to the anthropogenic land-use types used in ALCC scenario models and then trasnslated again in land-use harmonization schemes to produce quantitative estimates before being used for climate model simulations. The level of categorisation that is possible or necessary varies depending on the type of land use: it is clearly not useful to subdivide categories such as "no human land use" or "extensive/minimal land use". In the same way, there is no basis for subdividing some of the level 2 categories. For example, if there is "specialised fish production" it doesn't much matter what kind of fish are being farmed whereas if there is wet cultivation it does matter what type of crop is being grown and whether the wetland was natural or created for the purpose. We have already expanded this paragraph somewhat in response to comments by Almut Arneth, but we will further refine it to clarify the scheme as follows: Maps of the distribution of archaeological sites or of areas linked to a given food production system have been produced for individual site catchments or small regions (e.g. Zimmermann et al., 2009; Barton et al., 2010; Kay et al., in press). LandCover6k is developing global land-use maps for specific time windows, based on a global hierarchical classification of land-use categories (Morrison et al., 2018) based on land-use types that are widely recognised from the archaeological record. At the highest level, the maps distinguish between areas where there is no (or only limited) evidence of land use, and areas characterized by hunting/foraging/fishing activities, pastoralism, agriculture, and urban/extractive land use (Fig. 4). Except in the cases where land use is minimal (no human land use, extensive/minimal land use), further distinctions are subsequently made to encompass the diversity of land-use ac-

[Figure]

tivities in each land-use type (Fig. 4). A third level of distinction is made in the case of two categories (agroforestry, wet cultivation) where there are very different levels of intervention in different regions. Explanations of this terminology are given in Morrison et al. (2018). The LandCover6k land-use maps (see e.g. Fig. 5) will be based on different methods ranging from kernel-density estimates to expert knowledge depending on the quality and quantity of the archaeological information available from different regions.

Fig. 5 Too small legends. We will provide new figures to ensure that they are readable. Please see detailed explanations in the response to KjellstroÌLm.

Fig. 6 I don't understand the coupling between "LandCover 6k working group" and "HYDE 3.x". What does "→" mean? I don't understand many of the panels. What are the axes? What are the squares? What is the grey shading? A similar point was raised by KjellstroÌLm and we have expanded the text and modified the caption to explain this figure better

Fig. 7 Far too small legends. We will provide new figures to ensure that they are readable. Please see detailed explanations in the response to KjellstroÌLm.

Fig. 9 I don't understand this, but it seems to be more complicated than it sounds, but the surrounding text doesn't give much help. The text here describes the basis for using carbon cycle constraints on LULC. We will modify the caption to the Figure to clarify what this illustrative figure shows and so that it can be better understood in relation to the surrounding text, as follows: Illustration of the terrestrial C budget approach to evaluate LULC. The total terrestrial C balance (green circle 'total') is constrained by ice core records. The remainder (yellow slice 'remainder') is then calculated as the total terrestrial C balance (green circle 'total') minus the sum of separate estimates of natural components (blue slices 'Natural components') derived from modelling and/or upscaled observations. The remainder is effectively the emissions resulting from LULC changes, and can therefore be compared to LULC $CO_2$ emission estimates by carbon-cycle models.

Table 1 What does "Modern" mean here? If it is pre-industrial say so. If it is modern (= 20th century) explain why you don't use pre-industrial. This point has been raised by KjellstroÌLm and we have explained in that response that the PMIP protocol mandates modern geography and ice sheets for the pre-industrial simulation. We have expanded the text to explain this and modified the caption also.

Please also note the supplement to this comment:
https://www.geosci-model-dev-discuss.net/gmd-2019-125/gmd-2019-125-AC5-supplement.pdf

---

## Author Comment (AC7) · 28 Nov 2019

Response to comments by Erik KjellstroÌLm

Comments in italics, response in normal script, suggested changes to text in bold.

I'm not an expert in land-use or past changes in land-use but as a climate modeler with some limited experience in paleoclimate modelling I think that the paper would benefit from some more detailed discussion of potential limitations with the formulated strategy. In particular, the results illustrating the methods show: large spread, poor correlation and small differences between experiments with and without land-use (Figure

8). This could compromise the idea constraining land use change by climate model simulations. These plots show the direct comparison between gridded values of simulated mean annual temperature, mean temperature of the coldest month and mean annual precipitation and reconstructions as reconstructed from pollen data at these same gridcells. The spread is therefore not indicative of uncertainty, as suggested by the reviewer, but the geographic spread in climate across the region. The motivation for including anthropogenic land use in these experiments was the fact that there is a poor correlation between simulated and observed climate in the original experiment without land use changes. LULC was implemented using KK10. The plot shows that the correlation becomes slightly better for MAP but does not improve significantly for MAT and becomes worse for MTCO. We already know from comparisons with pollen data that the KK10 scenario is not "perfect" and this is our motivation for improving the scenario – so it would be hoped that the "improved" scenario leads to a better simulation of the climate. Certainly, if it does not lead to an improvement, then it will be meaningless to interpret the simulations as confirming the importance of LULC for correct simulation of climate during the Holocene. We have modified the caption to this figure in response to a specific comment (see below). We will modify the text describing this figure to clarify the expectations about the climate model tests, as follows: A second test of the realism of the improved LULC scenarios is to examine whether incorporating LULC changes improves the realism of the simulated climate when compared to palaeoclimate reconstructions (Figure 8). The mid-Holocene (6000 years ago, 6ka BP) is an ideal candidate for such a test because benchmark data sets of quantitative climate reconstructions are available (e.g. Bartlein et al., 2011), the interval has been a focus through multiple phases of PMIP and control simulations with no LULC have already been run, and evaluation of these simulations has identified regions where there are major discrepancies between simulated and observed climates e.g. the observed expansion of northern hemisphere monsoons, climate changes over Europe, the magnitude of high-latitude warming, and wetter conditions in central Eurasia (Mauri et al., 2014; Harrison et al., 2015; Bartlein et al., 2017). There are discernible anthropogenic

impacts on the landscape in many of these regions by 6 ka, although they are not as strong as during the later Holocene and they are not present everywhere. Nevertheless, the 6ka BP interval provides a good focus for testing improvements to the LULC scenarios. Such an evaluation would need to go beyond the global comparison made here (Figure 8) to regional comparisons to identify whether improvements in regions where there is a large anthropogenic impact on land cover do not result in a degradation in the simulated climate elsewhere.

In parallel to the climate model uncertainty, what is the uncertainty associated with the carbon cycle models proposed to be used for constraining the land use? Is it small enough to allow for a meaningful estimate of land use? I think the paper would benefit from a more in-depth discussion about these uncertainties. It is important to separate out the two applications of the carbon-cycle model simulations: first as a test of whether the scenarios are plausible and second as part of the transient Holocene climate simulations. In the offline simulations, we will use a single climate forcing but the intention is to use multiple carbon-cycle models - and this will allow us to evaluate the uncertainty associated with different models. This perhaps should have been made clearer. The planned transient model intercomparison further serves to address model uncertainty by design in using an ensemble of model simulations. This allows us to quantify model spread and therefore account for uncertainty related to differences between the models. However, the fact that planned simulations cover a very large temporal (∼12 kyr) and spatial (global) scale, restricts the possibilities to assess uncertainties in a more systematic way. In particular, with our activity, we do not aim at quantifying parametric model uncertainty because this would require a (very) large ensemble (on the order of thousands) of simulations with each individual model. This is not feasible. A single global model simulation covering 12 kyr takes on the order of weeks even for the fastest global models. We will expand the text describing the initial testing of the scenarios using carbon-cycle models to make it clearer that this is envisaged as a multi-model test, as follows: Transient simulations with a model that simulates $CO_2$ emissions in response to anthropogenic LULC can be used to test the

reliability of the LULC changes through time, by comparing results obtained with pre-scribed LULC changes through time against a baseline simulation without imposed LULC. Here we envisage using several different offline carbon-cycle models for this purpose in order to take account of uncertainties associated with inter-model differences. The carbon-cycle simulations will be driven by climate outputs (temperature, precipitation and cloud cover) from an existing transient climate simulation made with the ECHAM model (Fischer and Jungclaus, 2011) and $CO_2$ prescribed from ice-core records. The $CO_2$ emission estimates from these two simulations will then be evaluated using C budget constraints. This evaluation will allow us to pinpoint potential discrepancies between known terrestrial C balance changes and estimated LULC $CO_2$ emission in given periods over the Holocene.

Consideration could also be given if there would be a place for more detailed regional and local studies to further constrain land use? It is unclear what the reviewer is asking for here. The archaeological investigations are being carried out at a local scale and provide detailed regional records for some regions, which are then generalised for to continental scales. Both the detailed regional results and the continental maps will be used as inputs into the global LULC scenarios. The LULC scenarios necessarily have to be global for input into the climate model simulations. Similarly, the pollen-based constraints are site based and we have very detailed information on land use for some regions (e.g. Europe, North America) and less detailed information for others (e.g. tropics). Our evaluations will naturally make use of the detailed information where available.

General comments: Some words and concepts are quite difficult for a climate modeler (definition of time periods like the Holocene and Mesolithic and Neolithic times, taphonomic (L190)). The manuscript needs to be checked for consistency in how time is referenced (sometimes 6 ka BP, sometimes 6 ka). Also please explain what this means at the first reference. These points are raised below in the line-by-line specific comments, and our responses (and changes) are given there.

[Figure]

Line-by-line specific comments:

L1: Please don't use LULC in the title, better to spell out what it is about. We will change this to read: Development and testing of scenarios for implementing land use and land cover changes during the Holocene in Earth System Model experiments

L36: Unclear what is meant by "Current LULC scenarios". Is it current scenarios for the Holocene? Which part of the Holocene? Or, is it scenarios of LULC for the current climate (likely not, but it should be made more clear). We are referring to scenarios of LULC during the Holocene. We will clarify this as follows: Existing LULC scenarios of LULC changes during the Holocene are based on relatively simple assumptions and highly uncertain estimates of population changes through time.

L42-45: From this it is unclear if the paper is just on evaluation of scenarios or if it is also about further refinement of the scenarios. Our goal here is to provide a protocol for refining existing scenarios iteratively so that these scenarios can be used for climate model experiments. We realise that the abstract does not make this clear and will modify it as follows: In this paper, we document the types of archaeological data that are being collated and how they will be used to improve LULC reconstructions. Given the large methodological uncertainties involved, both in reconstructing LULC from the archaeological data and in implementing these reconstructions into global scenarios of LULC, we propose a protocol to evaluate the revised scenarios using independent pollen-based reconstructions of land cover and climate. Further evaluation of the revised scenarios involves carbon-cycle model simulations to determine whether the LULC reconstructions are consistent with constraints provided by ice-core records of CO2 evolution and modern-day LULC. Finally, the protocol outlines how the improved LULC reconstructions will be used in palaeoclimate simulations in the Palaeoclimate Modelling Intercomparison Project to quantify the magnitude of anthropogenic impacts on climate through time and ultimately to improve the realism of Holocene climate simulations.

L44: What kind of "carbon-cycle simulations" are referred to here? Earth-system model simulations? Carbon cycle model simulations? Anything else? We have modified the abstract (see above) to clarify this.

L53-54: The new IPCC special report on land states that 70% of land is being influenced by anthropogenic activities. Is there a discrepancy here? It is obviously difficult to provide an overall estimate of how much of the land surface is affected by human activities because it depends on whether the focus is on direct appropriation for agriculture resulting in a fundamental change in land cover or whether any anthropogenic influence is being taken into account. The Land Report states (section 1.1.2.2) that between 60–85% of the total forested area and between 72-89% of non-forested land is used, but it also makes it clear that the level of usage is variable with only 10% being intensively managed, two-thirds being moderately managed and the remainder at low intensities. Only about one third of the used land is associated with changed land cover. The Report states that differences in definitions and lack of information about management practice means that the estimates of human usage are uncertain. So, in this sense our statement is compatible with the Land Report, in that the estimated 40% refers to the area being used for agriculture and we go on to say that large parts of the rest of the land area are being influenced in some way by human activities. However, our point here is not to quantify the extent of use but simply to point out that there is considerable anthropogenic modification on the landscape globally. We will acknowledge the work of the Land Report – which came out after we submitted this paper – and modify this sentence as follows: Today, ca 10% the ice-free land surface is estimated to be intensively managed and much of the reminder is under less intense anthropogenic use or influenced by human activities (Arneth et al., 2019).

We will remove the following unnecessary references Foley et al., 2005 Ellis and Ramankutty, 2008 Ellis et al., 2010 Ellis et al., 2013 and add the reference to Arneth et al. (2019) Arneth et al., 2019. IPCC Special Report on Climate Change, Desertification, Land Degradation, Sustainable Land Management, Food Security, and Greenhouse

gas fluxes in Terrestrial Ecosystems.

L56-57: Please define what is meant by "Mesolithic and Neolithic". These archaeological periods are diachronous. The Mesolithic represents the final period of hunter-gather culture, and the Neolithic is associated with the emergence of agriculture, including domestication and more permanent settlements. We will modify the sentence to make this clearer for non-archaeologists as follows: Substantial transformations of natural ecosystems by humans began with the geographically diachronous shift from hunting and gathering characteristic of the Mesolithic to cultivation and more permanent settlement during the Neolithic period ....

L79: "LULC change during the Holocene". It is unclear what is meant here. Is it over the full Holocene? Or, from any particular time in early or mid Holocene to any point during late Holocene (preindustrial?). We agree that this is somewhat unclear. The experiments examine the impact of the change in 1850CE but this change represents the accumulated change in LULC through the Holocene. We will modify the text, as follows: At the global scale, the biogeophysical effects of the accumulated LULC change during the Holocene which resulted in reconstructed land cover patterns in 1850CE have been estimated to cause a slight cooling (0.17 °C) that is offset by the biogeochemical warming (0.9 °C), giving a net global warming (0.73 °C) (He et al., 2014).

L189: "lack of uniform sampling through time" – does this include different national sampling strategies/resources for archeological excavations/sampling? Most early archaeological sites represent occupation for only a limited period of time, although the same sites may be re-occupied at a later date. Differences in research traditions and foci in different regions means that particular periods may be intensively sampled and studied, while less interesting periods of time (from an archaeological perspective) are neglected. Lack of resources and preservation issues means that it is virtually impossible to obtain a uniform sampling of archaeological records in space and time and in any case such a sampling does not currently exist for most regions. In response to a slightly different comment by Almut Arneth, we propose to modify this sentence as follows: There are biases that could affect the expected one-to-one relationship between number of people and number of radiocarbon dates on archaeological material, including lack of uniform sampling through time and space caused by different archaeological research interests and traditions in different regions) and increased preservation issues with increasing age.

L190: What is taphonomic? Taphonomic processes are those which result in post-deposition modification of deposits, here including decomposition or erosion. Here we simply meant to say that there is a loss of information because preservation becomes less reliable with age. We have modified the sentence (see above) to remove the jargon.

L331-343: Here, it is unclear whether the "already produced reconstructions" are products of REVEALS or if there are any other methods that have been involved. These reconstructions, which are illustrated in Figure 7, were made using REVEALs. We will clarify this and also include an additional references to the figure at this point, as follows: LandCover6k has already produced reconstructions using REVEALS for the northern extratropics (see e.g. Figure 7). These reconstructions provide snapshots through the Holocene the Holocene with a time resolution of 500 years until 0.7ka BP, and three historical time windows (modern–0.1ka BP, 0.1–0.35ka BP, and 0.35–0.7ka BP).

L361: Suggest changing "observed climate" to "reconstructed climate". We will make this change (actually L357)

L386-390: Here it is discussed changes in land use over time. The text gives the impression that there is always increasing land use with time "more conversion in earlier periods implies less conversion in later periods". Seems logical, but does this argument hold in a situation when land use is fluctuating with time (e.g. no land use – some land use – forest regrowth – no land use – again more land use . . .)? We are not implying that land use always increases through time, because indeed the archaeological

evidence shows that this is not the case and this is illustrated in Figure 5 for example. What we are trying to explain is that the cumulated amount of land converted to agriculture during the Holocene must sum to the amount of agricultural land today. So, if there is a lot of conversion early on, then there must either be less later or large parts of the converted land must have reverted to non-agricultural land. We will try to make this clearer by modifying the text. as follows: First, reconstructions of the total land under agricultural use must converge on the present-day state, which is relatively well constrained by satellite land-cover observations and national statistics on the amount of land under use. Reconstructing the extent of past LULC thus reduces to allocating a fixed total amount of land conversion from natural to agricultural use over time. More conversion in earlier periods implies either abandonment of agricultural land or less conversion in later periods.

L395: "to" missing after "due". We will correct this (actually line 390)

L440: How is land-use implemented in the models? Is it binary (i.e. 0 or 1) or fractional? In the latter case I guess that dynamical vegetation models could be used in combination with the land use information to derive vegetation type for the part of a gridbox not associated with land use. Land use is currently not implemented in the mid-Holocene simulations. The implementation in the CMIP6 past1000 and historic simulations varies with the model; most of the models use fractional coverage. Not all of the models include dynamic vegetation, or rather have dynamic vegetation "switched on" in their piControl experiment, but for those that do we are indeed proposing that the vegetation is simulated in that fraction of a gridcell that is not affected by LULC. We will revise the paragraph describing the mid-Holocene simulations to make this clearer, as follows: The mid-Holocene (and its corresponding piControl) is one of the PMIP entry cards in the CMIP6-PMIP4 experiments (Kageyama et al., 2018; Otto-Bliesner et al., 2017) and it is therefore logical to propose this period for LULC simulations. The LULC sensitivity experiment (midHoloceneLULC) should therefore follow the CMIP6-PMIP4 protocol, that is it should be run with the same model components and following the

same protocols for implementing external forcings as used in the two CMIP6-PMIP4 experiments (Table 1). Thus, if the piControl and midHolocene simulations are being run with interactive (dynamic) vegetation, then the midHoloceneLULC experiment should also be run with dynamic vegetation in regions where there is no LULC change. For most models, this means that the LULC forcing is imposed as a fraction of the grid cell and the remaining fraction of the grid cell has simulated natural vegetation.

L444-445: "free atmospheric CO2" needs a better explanation – for instance something like "..., allowing atmospheric CO2 concentrations to evolve in concert with fluxes to and from land and oceans". We will change this to: Thus, modelling groups who are running the midHolocene experiment with a fully interactive carbon cycle could also run the LULC experiment allowing atmospheric CO2 to evolve interactively, subject to the simulated ocean and land C balance.

L466: Please elaborate a bit on how good the assumption on "equilibrium" is for the Mid-Holocene? Was the carbon cycle (and climate) at equilibrium at that time? In the text, we are referring to starting the transient experiments from the mid-Holocene experiment because these equilibrium experiments are mandated to have a long enough spin-up to be in equilibrium before the experiment is run (see Otto-Bliesner et al., 2017). Whether the carbon cycle and climate was at equilibrium in the real world is not an issue. In the present context, where we address LULC CO2 emissions that evolve over centuries to millennia, disequilibrium effects are relatively small. This is due to the much shorter time scale of emissions occurring after forest clearance (on the order of years to decades). The longer time scales of forest regrowth (centuries) might be relevant too, where agricultural land abandonment and forest regrowth are important. We will clarify the issue of the mid-Holocene experimental equilibrium in the protocol, as follows: We suggest that this transient simulation (holotrans) should start from the pre-existing midHolocene simulation to capitalise on the fact that the midHolocene simulation have been spun up for sufficiently long (Otto-Bleisner et al., 2017) to ensure that the ocean and land carbon cycle is in equilibrium at the start of the transient experiment

(Table 2).

L482-488: All references here are more than 10 years old. Are there no more recent studies of relevance? Unfortunately, there are no more recent continental scale reconstructions of climate through the Holocene – although there are ongoing projects that are planning to revisit these reconstructions for Europe and North America taking advantage of more extensive pollen data sets and newer reconstruction techniques. There are newer reconstructions for Europe and the USA for individual sites, but site-based model evaluation is difficult and here we only give references to individual sites in regions where there are no continental-scale reconstructions. We could add more references to reconstructions at individual sites but perhaps it would be better to clarify why such data is not particularly helpful for model evaluation, as follows: Quantitative climate reconstructions through the Holocene at a regional scale are currently only available for Europe (Davis et al., 2003) and North America (Viau et al, 2006; Viau and Gajewski, 2009). There are time series reconstructions for individual sites outside these two regions (e.g. Nakagawa et al., 2002; Wilmshurst et al., 2007; Ortega-Rosas et al., 2008), but it is difficult to rely on such reconstructions for model evaluation because of the differences in resolution between the models and the geographic scale sampled by individual sites. However, the simulated time-course of $CO_2$ emissions can be compared to the ice core records.

Figure 1: The color scale with the relatively dark green makes it difficult to see any of the rather small areas with land-use. It is difficult to understand why these two years have been chosen from the datasets (why not use the same reference year?). The font size at the color bar is too small. The two data sets (KK10, HYDE3.2) do not have outputs for every year and so we have chosen the two available intervals that correspond most closely to the mid-Holocene time interval from each. They are 50 years apart, which given the uncertainties on radiocarbon dating of this time interval can be considered indistinguishable from one another. We will redraw this figure (and the other figures) to ensure that the font size is readable throughout.

Figure 2: The figure is difficult to read and it is not easy to see what is the final outcome of the scheme based on the figure. If it is something like "LULC scenario" I guess this should be something popping out on the right-hand side after going through the three steps in Phases 1-3. Also, it is not clear from the figure if there is any iterative part in the process where info is added to the scenarios based on constraints from phases 2-3? This could be better explained here and would also help to make the paper a bit more clear on a general level. We have expanded our description of this Figure and the general protocol in response to comments by Almut Arneth. We will redraw this figure (and the other figures) to improve readability.

Figure 3. Here, font sizes are too small everywhere. What is SDPs? Please explain what the shading is for the maps (areas under human use?) and give a color bar. What are the circles in the lowermost panels? In addition to revising the figure to improve readability, we will change the to explain the abbreviation SDP and the shading, as follows: Reconstruction of changes in population size in the Iberian Peninsula during the Holocene (9000 to 2000 BP, 9ka to 2ka BP) using summed probability distributions (SPDs) of radiocarbon dates (data after Balsera et al., 2015). The red line indicates the onset of agriculture in the region. The lower panels show areas under human use at 6ka (left) and 4ka (right) using kernel density estimates, where the white dots are actual archaeological sites and the shading shows the implied density of occupation.

Figure 4. Here is a box (Extensive/Minimal land use) that lacks some Level 2/3 information. Or it is redundant and can be removed? The labels on the land-use classes are quite specialized and several of the words are not everyday terms from my perspective (pastoralism, chinampas, taro pondfields, Peri-urban, Swidden). It would be good if these were a bit better explained, alternatively use different words). Also, why are there only Level 3 boxes for some of the Level 2 boxes? The Figure is included for illustrative purposes and shows the scheme of land-use categories developed by LandCover6k to be used by the archaeological community to map land-use in different regions of the world. The terminology is that used to describe different kinds of
agriculture by archaeologists, and there is a handbook (which we can refer to) that defines these terms. As we explain in the text, these land-use types will have to be translated to the anthropogenic land-use types used in ALCC scenario models and then trasnslated again in land-use harmonization schemes to produce quantitative estimates before being used for climate model simulations. The level of categorisation that is possible or necessary varies depending on the type of land use: it is clearly not useful to subdivide categories such as "no human land use" or "extensive/minimal land use". In the same way, there is no basis for subdividing some of the level 2 categories. For example, if there is "specialised fish production" it doesn't much matter what kind of fish are being farmed whereas if there is wet cultivation it does matter what type of crop is being grown and whether the wetland was natural or created for the purpose. We have already expanded this paragraph somewhat in response to comments by Almut Arneth, but we will further refine it to clarify the scheme as follows: Maps of the distribution of archaeological sites or of areas linked to a given food production system have been produced for individual site catchments or small regions (e.g. Zimmermann et al., 2009; Barton et al., 2010; Kay et al., in press). LandCover6k is developing global land-use maps for specific time windows, based on a global hierarchical classification of land-use categories (Morrison et al., 2018) based on land-use types that are widely recognised from the archaeological record. At the highest level, the maps distinguish between areas where there is no (or only limited) evidence of land use, and areas characterized by hunting/foraging/fishing activities, pastoralism, agriculture, and urban/extractive land use (Fig. 4). Except in the cases where land use is minimal (no human land use, extensive/minimal land use), further distinctions are subsequently made to encompass the diversity of land-use activities in each land-use type (Fig. 4). A third level of distinction is made in the case of two categories (agroforestry, wet cultivation) where there are very different levels of intervention in different regions. Explanations of this terminology are given in Morrison et al. (2018). The LandCover6k land-use maps (see e.g. Fig. 5) will be based on different methods ranging from kernel-density estimates to expert knowledge depending on the quality and

quantity of the archaeological information available from different regions.

Figure 5. This figure is not easily readable. The font size in the legends is way too small, the red dots in the upper panels are hardly distinguishable and the land-cover classes in the lowermost figure are not readable. Is the order left/right OK here? The figure indicates more people and land use at the earlier period (right panels) if I'm interpreting the figures correctly. In the figure caption "cal BC and BP" are used without definition anywhere. Also in the figure caption intervals defining the Middle and Early Neolithic time periods are given. Are these related to the more general statement on l56/57? We will redraw all the figures to make them more readable. Indeed the figure does show that there were more people during the earlier period than the later period, and this is one of the reasons we chose this as an illustration to make the point that the impact of human activities is not unidirectional! The more general statement does not imply that the changes are unidirectional, as we have now clarified (see above). We realise that there are inconsistencies in the way time is expressed in the figures and figures captions (we do not refer to specific times in the text). We would like to keep both BP and BCE dates because the former terminology is used by climate modellers and the Quaternary geology community, and the latter by archaeologists. However, we will define the terms consistently in each of the captions, as follows:

Figure 1: Land use at ca 6000 years ago (6ka BP, 4000 years BCE) from the two widely used global historical land-use scenarios HYDE 3.2 (top panel, Klein Goldewijk et al. 2017a) and KK10 (bottom panel, Kaplan et al. 2011), illustrating the large disagreement between LULC scenarios at a regional scale. In both scenarios, the land-sea mask and lake areas are for the present day.

Figure 3: Reconstruction of changes in population size in the Iberian Peninsula during the Holocene (9000 years to 2000 years ago, 9ka BP to 2ka BP) using summed probability distributions (SPDs) of radiocarbon dates (data after Balsera et al., 2015). The red line indicates the onset of agriculture in the region. The lower panels show areas under human use at 6ka BP (left) and 4ka BP (right) using kernel density estimates,

where the white dots are actual archaeological sites and the shading shows the implied density of occupation.

Figure 5: An example of regional land-use mapping. The plots show the distribution of archaeological sites superimposed on kernel density estimates of the extent of land-use based on the density of sites (top panels), and superimposed on the LandCover6ka land-use classes (bottom panels) for the Middle Neolithic (3600-3400 years BCE, 5600-5400 years BP, 5.6-5.4 ka BP) (left panels) and the Early Neolithic (3750-3600 years BCE, 5750-5600 years BP, 5.7-5.6 ka BP) (right panels) of Ireland. Data points derive from 14C dated archaeological sites and distributions of settlements and monuments that have been assigned to each archaeological period following the dataset published in McLaughlin et al. (2016). In areas characterized by low-level food production, agricultural land (crop growing and grazing, combined) probably occupies between 10-15% of the total grid cell area in eastern and western coastal areas, whilst inland agricultural land likely represent 5% or less of the total grid cell area.

Figure 7: Northern extratropical (>40°N) mean fractional cover of open land at 6000 years ago (6ka BP: left panel) and 200 years ago (0.2ka BP: centre panel) estimated using REVEALS, and the difference in fractional cover between the two periods (right panel), where red indicates an increase in open land and blue a decrease (after Dawson et al., 2018).

Figure 6. Realizing that these figures are conceptual, but they still need some better illustration. What are the different "squares" in the left panel second from the top? Grid squares on a spatial map? Same question for the plots on the third row (and what is the bar with shading representing?)? Units lowermost left panel? Why is there a label "HYDE 3.x" on the top? We have already modified the caption to this figure in response to comments from Almut Arneth (see below) to explain more clearly what this illustrative figure is about. Schematic illustration of the proposed implementation of 14C-based population estimates, date of first agriculture, land-use maps, and land-use per capita information in the HYDE model (here indicated as HYDE3.x). The archaeological data

are represented as values for a grid cell in geographic space at a given time for date of first agriculture and land use, but as a time series for a specific grid cell for population and land-use per capita. In the case of population estimates, date of first agriculture and land-use per capita data, we show the initial estimate and the revised estimate after taking the archaeological information into account in the HYDE3.x plot. It should be assumed in the case of the land-use mapping that the original estimate was that there was no land use in this region.

Figure 7. A suggestion here could be to remove the panel with the differences and make the other two a bit bigger and more easy to read (including larger font size on the color bar). We will replot this figure to make it clearer.

Figure 8. What are all the dots in the panels? Are the sites covering large areas? Biased to some regions? Evenly spread? Are all three panels for areas north of 30N? What are the associated uncertainty bars with the proxy-based data? With the models? The dots represent the individual grid cells where comparisons are possible. The Bartlein et al data set is a gridded data set derived from site-based pollen-based reconstructions. The original sites are certainly not evenly spread and there are more grids in some regions than others. All this information is given in the Bartlein et al. paper from which these data are sourced. As it says in the caption, all of the plots are for the region north of 30° N, and this region was chosen because it has the most even coverage. We do not show uncertainty bars here, either for the model or for the data. What we show is the strength of the relationship between the observations and the simulations in the two experiments. Nevertheless, we will expand the caption to make it clearer what this comparison involves, as follows: Figure 8: Quantitative comparison of the change in climate between the mid-Holocene (6ka) and the pre-industrial period as shown by pollen-based reconstructions gridded to 2 x 2° resolution to be compatible with the model resolution (from Bartlein et al., 2011) and in simulations with and without the incorporation of land-use change (from Smith et al., 2016). The imposed land-use changes at 6000 years ago (6ka BP) were derived from the KK10 scenario

(Kaplan et al., 2011). The plots show comparisons of mean annual temperature (MAT), mean temperature of the coldest month (MTCO) and mean annual precipitation (MAP) for the northern extratropics (north of 30° N), where each dot represents a model grid cell where comparisons with the pollen-based reconstructions is possible. Although the incorporation of land use produces somewhat warmer and wetter climates in these simulations, overall the incorporation of land-use produces no improvement of the simulated climates at sites with pollen-based reconstructions.

Comments on Table 1: Why is "Modern" paleogeography and ice sheets used instead of "piControl"? And, how (if at all?) are these two differing? In the table "LC6k" is used supposedly for "LandCover6k", please spell out. What does it mean that pasture and crop distributions are "imposed"? I guess "imposed on top of the default vegetation in the 6ka experiment". These simulations follow the standard PMIP protocol for the mid-Holocene simulation as described by Otto-Bleisner et al. (2017). We say this in the text. These mid-Holocene simulations make no change in geography (land-sea distribution and topography) or ice sheet extent, i.e. they prescribe modern values for these. In point of fact, the real-world difference in these two things between the modern day and the pre-industrial (1850 CE) is negligible and not distinguishable at the model resolution. We will change the description of the imposition of crop and pasture in the table to read: pasture and crop distribution prescribed from the revised scenario We will also change the caption to clarify the relationship with the PMIP simulations, as follows: Boundary conditions for CMIP6-PMIP4 and the mid-Holocene LULC experiments. The boundary conditions for the CMIP6-PMIP4 piControl and midHolocene are described in Otto-Bleisner et al. (2017) and are given here for completeness.

Please also note the supplement to this comment:
https://www.geosci-model-dev-discuss.net/gmd-2019-125/gmd-2019-125-AC7-supplement.pdf